# Acoustic-driven magnetic skyrmion motion

Yang Yang [1,6], Le Zhao [2,6], Di Yi[3,6], Teng Xu[2], Yahong Chai[1], Chenye Zhang[1], Dingsong Jiang[1], Yahui Ji[1], Dazhi Hou [4,5], Wanjun Jiang [2]✉, Jianshi Tang [1], Pu Yu [2], Huaqiang Wu [1] & Tianxiang Nan [1]✉

Magnetic skyrmions have great potential for developing novel spintronic devices. The electrical manipulation of skyrmions has mainly relied on current-induced spin-orbit torques. Recently, it was suggested that the skyrmions could be more efficiently manipulated by surface acoustic waves (SAWs), an elastic wave that can couple with magnetic moment via the magnetoelastic effect. Here, by designing on-chip piezoelectric transducers that produce propagating SAW pulses, we experimentally demonstrate the directional motion of Néel-type skyrmions in Ta/CoFeB/MgO/Ta multilayers. We find that the shear horizontal wave effectively drives the motion of skyrmions, whereas the elastic wave with longitudinal and shear vertical displacements (Rayleigh wave) cannot produce the motion of skyrmions. A longitudinal motion along the SAW propagation direction and a transverse motion due to topological charge are simultaneously observed and further confirmed by our micro-magnetic simulations. This work demonstrates that acoustic waves could be another promising approach for manipulating skyrmions, which could offer new opportunities for ultra-low power skyrmionics.

Using magnetic skyrmions, the particle-like spin textures, as controllable information carriers offer potentials for high density and low power spintronic memory and logic applications[1–8]. To develop skyrmion-based devices, e.g. skyrmion racetrack memory, efficient manipulation of skyrmions is crucial. Electric current manipulation of skyrmions has been previously demonstrated in asymmetric magnetic multilayers by means of current-induced spin-orbit torques or thermal gradients, among many others[9–20]. On the other hand, the electric-field control of skyrmions via magnetoelectric or magnetoelastic effect could provide more energy-efficient approaches with an extremely low Joule heating and hence low power consumption[21,22]. Such control means can be achieved, for example, by the static strain modification of magnetic anisotropy and Dzyaloshinskii-Moriya (DM) interaction[23–25], or by dynamic strains using acoustic waves through strong magnon-phonon coupling[26,27]. In particular, surface acoustic waves (SAWs) are long-range carriers (wave propagating over milli-

meter distances through ferromagnets) for dynamic strains[28–36], which have been used as an efficient source for generating skyrmions by the SAW-induced spatiotemporally varying strains and inhomogeneous effective torques[37]. A recent theoretical model also suggested the skyrmion motion driven by counter-propagating SAWs[38]. Yet the electric-field induced static strain or acoustic wave control of the skyrmion motion has not been experimentally demonstrated. Although SAWs have been used as a noncontact and controllable method to manipulate nano/microparticles[39,40], electrons[41], and qubits[42], the associated efficiency seems not high enough to induce the motion of magnetic skyrmions.

Here, we experimentally study the SAW-driven directional motions of Néel-type magnetic skyrmions due to the strong magnetoelastic coupling in magnetic multilayers integrated with on-chip piezoelectric transducers. By controlling the relative orientation between the propagation of acoustic wave and crystal orientation of

[1]School of Integrated Circuits and Beijing National Research Center for Information Science and Technology (BNRist), Tsinghua University, Beijing, China. [2]Department of Physics, Tsinghua University, Beijing, China. [3]School of Materials Science and Engineering, Tsinghua University, Beijing, China. [4]ICQD, Hefei National Laboratory for Physical Sciences at Microscale, University of Science and Technology of China, Hefei, Anhui, China. [5]Department of Physics, University of Science and Technology of China, Hefei, Anhui, China. [6]These authors contributed equally: Yang Yang, Le Zhao, Di Yi. ✉e-mail: jiang_lab@tsinghua.edu.cn; nantianxiang@mail.tsinghua.edu.cn

the piezoelectric materials, we generate both Rayleigh waves (with shear vertical and longitudinal displacements) and shear horizontal (SH) waves (with only shear horizontal displacements) that can be applied to skyrmions at the same sample area. We find that the Rayleigh wave can generate but not move the skyrmions due to its dominant vertical displacement (Fig. 1a), which is consistent with the early report[37]. By contrast, the SH wave can efficiently move skyrmions as a result of the strong magnetoelastic coupling induced by the in-plane strain gradients. The observed directional motion shows a longitudinal component along the wave propagation direction, and transverse components with their signs depending on the topological charges, in analogy to the skyrmion Hall effect[12,43]. These experimental observations are further confirmed by our micromagnetic simulations. Our results not only provide an efficient approach to drive the skyrmion motion by electric field-induced strain wave, but also demonstrates the SAW alone could serve as another versatile platform to explore the skyrmion dynamics.

## Results

The difference of the skyrmion motion driven by Rayleigh and SH waves can be captured by the micromagnetic simulations, which can be done by considering magnetoelastic coupling, exchange coupling and DM interaction (see Method and Supplementary Information note 2 for detail). Figures 1b and 1e show the simulated spatial distribution of the normalized out-of-plane magnetization component $m_z$, magnetoelastic energy density and total energy density for skyrmions that were driven by Rayleigh and SH waves, respectively. We observe that a skyrmion does not exhibit significant movement under the action of a

Rayleigh wave. The amplitude of the shear vertical displacement is usually larger than that of the longitudinal displacement in a Rayleigh wave. The magnetoelastic energy density distribution of a skyrmion strongly depends on the relative size of the skyrmion in comparison to the wavelength of the SAW. When the size of the skyrmion is comparable with the wavelength of a SAW, the strain gradient induced by a SAW becomes non-uniform across the skyrmion, resulting in an asymmetric magnetoelastic energy density distribution. The magnetoelastic energy density distribution of a skyrmion under a shear vertical wave shows a left-right asymmetry that is different from that under an SH wave including both left-right and up-down asymmetry due to the different displacement modes. The total energy density of the skyrmion (including the magnetoelastic energy, anisotropy energy, magnetostatic energy, exchange and DM energy) illustrates a symmetric distribution under a shear vertical wave, while an asymmetric distribution along the diagonal axis under an SH wave which causes the skyrmion to move towards the lower energy density direction. When we set the wavelength of a SAW much larger than the diameter of the skyrmion, the strain gradient induced by a SAW can be considered to be uniform across the skyrmion. In this case, the magnetoelastic force density distribution of a skyrmion is nearly symmetric. And the net magnetoelastic force on a skyrmion approaches zero. This has also been predicted by the analytical model in the early report[38].

We study the skyrmions motion driven by SAWs in Ta (5 nm)/ $Co_{20}Fe_{60}B_{20}$ (CoFeB, 1 nm)/MgO (1 nm)/Ta (2 nm) multilayers, because the multilayers show a weaker pinning effect than that in Pt/Co/Ta multilayers[10,20]. This is due to the fact that the amorphous CoFeB has a

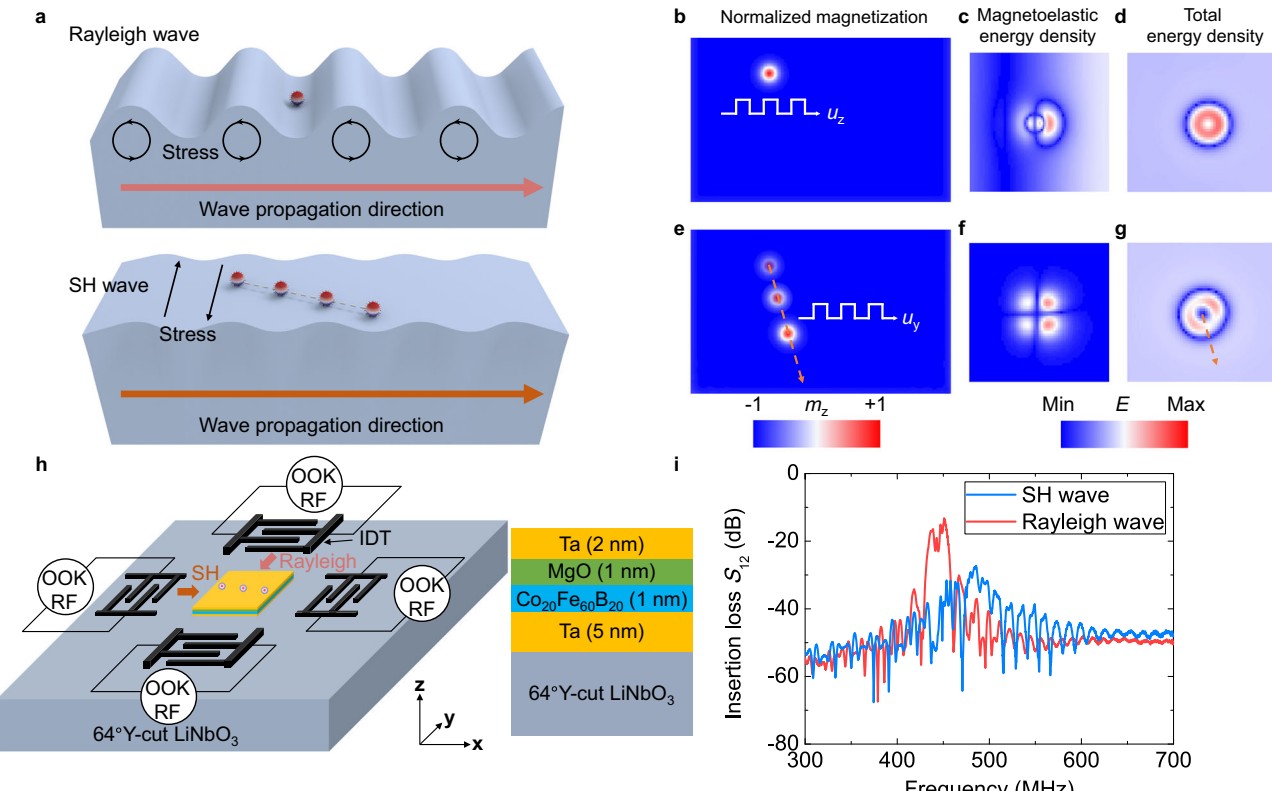

**Fig. 1 | Concept of skyrmion motion driven by SAWs and the device schematics.** **a** Schematic diagram of skyrmion dynamics driven by a Rayleigh wave or a shear horizontal (SH) wave. **b** Simulated skyrmion pinning under an elastic wave with periodic shear vertical displacements. Due to the limitation of the computational capacity, the diameter of the simulated skyrmion is set as 30 nm. The wavelength is 240 nm which is eight times as large as the simulated skyrmion size. The color scale represents the normalized (out-of-plane) magnetization component $m_z$. **c** the magnetoelastic energy density, **d** The total energy density of the skyrmion under an elastic wave with periodic shear vertical displacements. **e** Simulated skyrmion motion driven by an SH wave. The arrow denotes the trajectory of motion. **f** The magnetoelastic energy density, **g** the total energy density of the skyrmion under an SH wave. **h** Schematic of the SAW delay line device configuration. **i** Transmission spectrums ($S_{12}$) of the SH wave and the Rayleigh wave. The wavelength of SAWs is 8 μm.

strong magnetoelastic coupling and a low damping parameter[44]. The magnetic properties of the multilayers on a LiNbO₃ substrate were characterized by a polar magneto-optic Kerr effect (MOKE) magnetometry (Supplementary Fig. S1a). The average diameter of skyrmions generated by magnetic field pulses is estimated to be around 1 μm (Supplementary Fig. S1c). The multilayers and interdigital transducers (IDTs) were integrated on a 64°Y-cut LiNbO₃ piezoelectric substrate, as shown in Fig. 1h (optical image of the fabricated devices is shown in Supplementary Fig. S1d). By controlling the angle between the SAW propagation direction and the orientation of the piezoelectric substrate, the piezoelectric constant matrix can be transformed (see Supplementary Information Note 2), and thus the SH wave or Rayleigh wave can be generated independently with the wave propagation along the **x** or **y** direction, respectively (Fig. 1d). Figure 1i shows a transmission spectrum ($S_{12}$) between two IDTs, which is obtained by using a vector network analyzer where the resonance frequencies of the SH wave mode and the Rayleigh wave mode are 486 MHz and 451 MHz, respectively. The propagation attenuation of the SH wave mode in the 64°Y-cut LiNbO₃ crystal is larger than that of the Rayleigh wave mode, resulting in a lower resonant peak for the SH wave compared to the Rayleigh wave.

We first study skyrmion generation by using SAWs. Figure 2 shows the MOKE images for the evolution of magnetic textures. At a zero magnetic field, maze domains are observed (Fig. 2a). Then we start with a state with almost no magnetic texture by eliminating the initial maze domain structure (by applying out-of-plane magnetic field of −0.8 mT), as shown in Fig. 2b. Magnetic skyrmions with a topological charge $Q = +1$ are created after exciting a propagating Rayleigh wave or SH wave with a pulse duration of 300 ms at the resonance frequencies, as shown in Figs. 2c and 2d. The positive or negative sign of $Q$ represents the center magnetization of the skyrmion being up or down. Fig. 2e and f show the

evolution of skyrmion densities and sizes as a function of applied RF powers, which are created by SH and Rayleigh waves. Note that the skyrmion density created by a Rayleigh wave is slightly smaller than that in the previous report[37], which could be attributed to the shorter pulse duration in our experiments. We find that the SH wave can generate skyrmions more efficiently than the Rayleigh wave when the RF power is above 20 dBm. This indicates that the SH wave mode can couple with skyrmions more effectively because the SH wave with its dominant in-plane shear horizontal displacement produces a stronger in-plane magnetoelastic energy. The average sizes of skyrmions generated by both SH and Rayleigh waves are estimated to be around 1 μm, similar to that generated by the magnetic field.

We then study the skyrmion motion by applying continuous SAW pulses with a fixed RF power of 26 dBm. Figure 3a–d (e-h) illustrate MOKE images of $Q = +1$ ($Q = −1$) skyrmion after exciting 1st– 4th SH wave pulse with a duration of 300 ms (also see Supplementary movies 1 and 2). We find that the skyrmions not only move along the wave propagation direction (**x** axis), but also exhibit a transverse velocity component (**y** axis), in analogy to the skyrmion Hall effect[12]. The motion distances $d$ ($d = \sqrt{d_x^2 + d_y^2}$) of the circled skyrmions ($Q = \pm 1$) are around 3 μm after each pulse. The SH wave with a wavelength of 10 μm also moves skyrmions (Supplementary Fig. S7). Statistically, 32% of the skyrmion population in the whole sample shows motion distance $d > 1$ μm (Supplementary Fig. S8). Note that the motion distances can be further improved by increasing RF powers or decreasing the wavelength of SAWs (Supplementary Fig. S8). By contrast, we do not observe any skyrmion motion driven by Rayleigh waves although the power of the receiving IDTs for a Rayleigh wave is 12 dBm higher than that of an SH wave (Supplementary Fig. S9), which is consistent with our micromagnetic simulations.

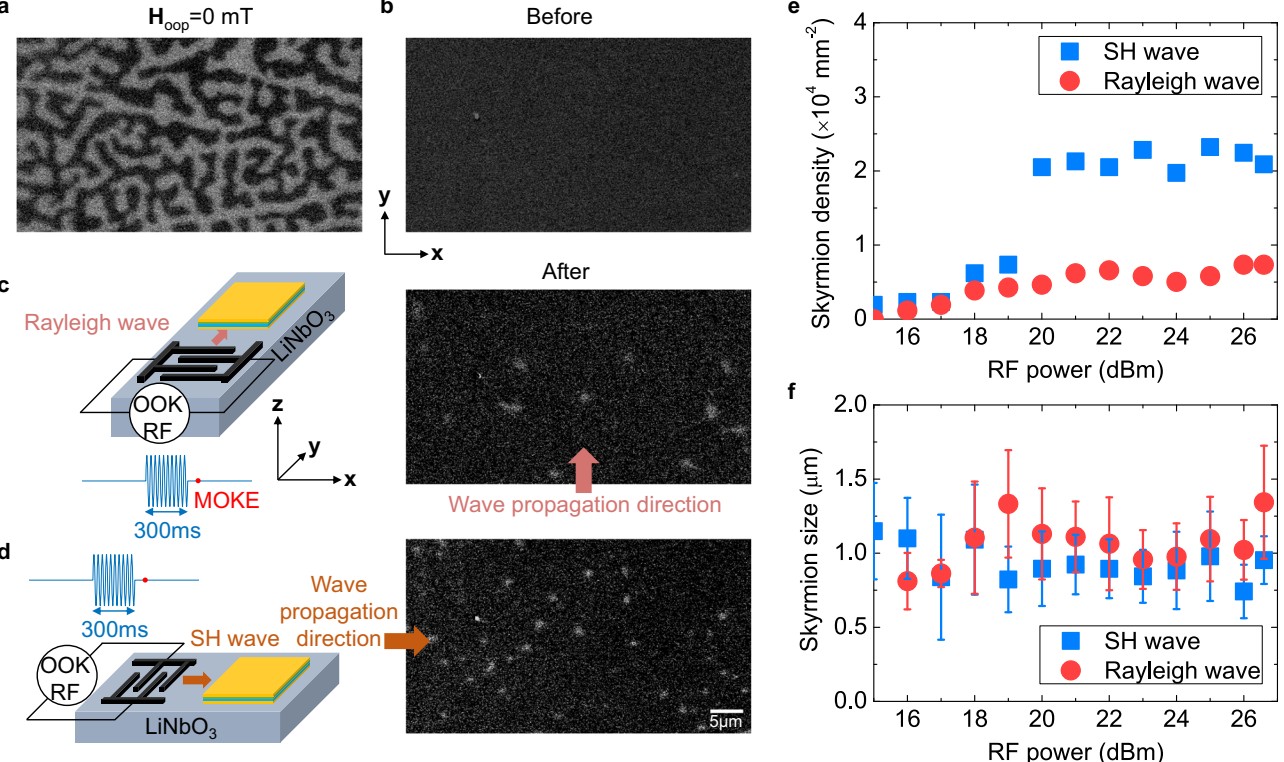

**Fig. 2 | Generation of skyrmions by Rayleigh and SH waves.** Polar MOKE images (**a**) of the maze domain at a zero magnetic field, (**b**) before exciting a propagating SAW wave, $H_{oop} = −0.8$ mT, the magnitude of the applied out-of-plane field increases over the negative saturation field and then decreases to −0.8 mT, (**c**) after exciting a propagating Rayleigh wave, (**d**) after exciting a propagating SH wave. The RF power is 26 dBm. The SAW pulse duration is 300 ms. The scale bar is 5 μm. **e** Densities of skyrmions created by SH waves and Rayleigh waves as a function of RF powers. **f** Skyrmion sizes induced by SH waves and Rayleigh waves as a function of RF powers. The error bars correspond to the standard deviation.

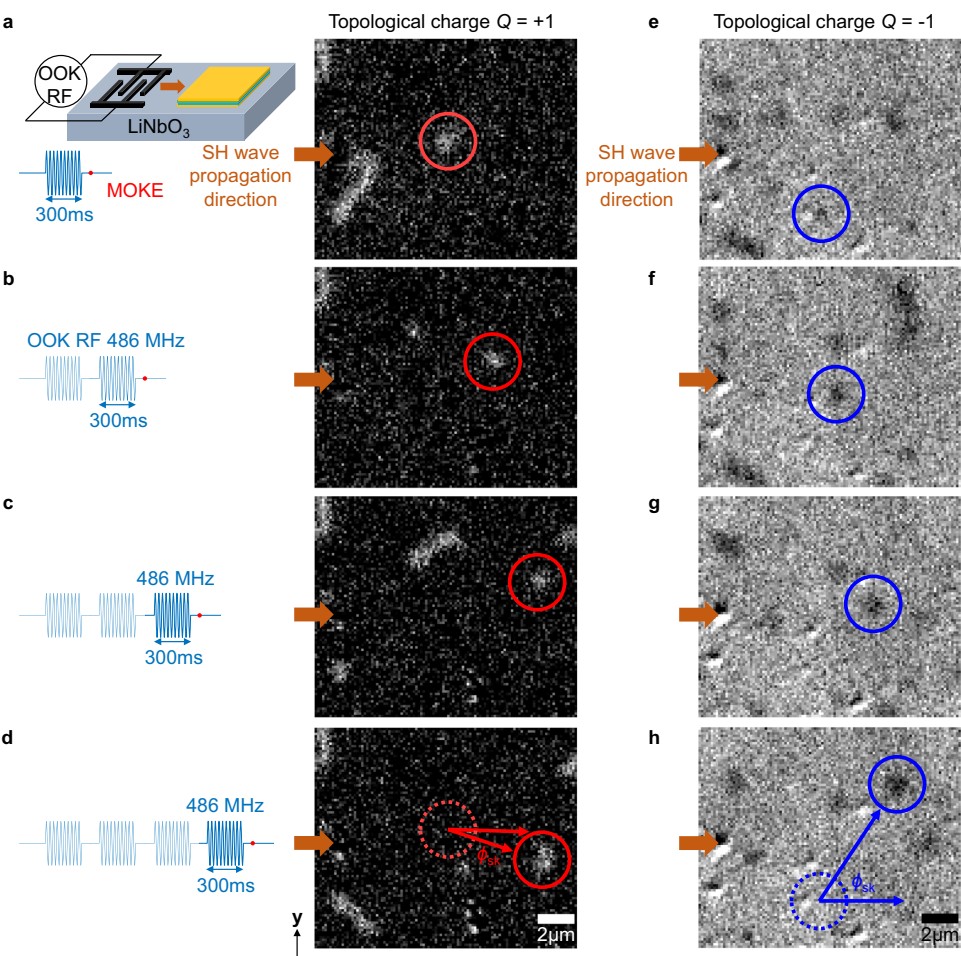

**Fig. 3 | Skyrmion motions driven by SH waves.** The MOKE images of (**a**–**d**) $Q = +1$ (**e**–**h**) $Q = -1$ skyrmions motion after the exciting of 300 ms SH wave pulses. The wavelength of SAWs is 8 μm. The RF power applied to IDTs is 26 dBm. The SH wave propagating direction is from left to right. The SAW creates skyrmions with a topological charge $Q = \pm1$ when the positive or negative out-of-plane magnetic field ($\mathbf{H}_{oop} = \pm0.8$ mT) is applied. The scale bar in the MOKE images is 2 μm.

## Discussion

We estimate the skyrmion velocity driven by SH wave to be about 10 μm/s, which is similar to that driven by current-induced spin-orbit torques with small current densities[43][45]. This could suggest that the skyrmion motion driven by SH waves under the present experimental condition remains in the creep regime. By progressively increasing the RF power, it is possible to increase the skyrmion velocity, but the wave amplitude is saturated (Supplementary Fig. S9). To transform from the creep regime to the flow regime with a much higher skyrmion velocity, one can use magnetic films with stronger magnetoelastic coupling constants and low damping parameters (Supplementary Information Note 3).

We summarize the motion trajectories (both $d_x$ and $d_y$) of 18 different skyrmions in Fig. 4a. In the creep regime, the skyrmion is easily trapped by the randomly distributed pinning sites. Nonetheless, the skyrmions with $Q = +1$ ($Q = -1$) move consistently along the wave propagation direction with $d_y < 0$ ($d_y > 0$). This is in agreement with our micromagnetic simulation (Figs. 4b and 4c). The average deflection angles ($\phi_{sk} = \arctan(d_y/d_x)$) of $Q = -1$ and $Q = +1$ skyrmions are around 49.5° ± 15.2° and −34.2° ± 17.7°, respectively. The large variation of deflection angles can be attributed to the pinning potential induced by random defects. By solving the Thiele equation[46], our analytical calculation reveals that the deflection angle is determined by the damping parameter and the ratio of effective magnetoelastic forces along the **x** and **y** axes ($F_x$ and $F_y$). In Fig. 4d, the solid curves show the calculated deflection angle as a function of damping parameters with

$F_y/F_x = 0.3$ (reasonable value for an SH wave). The calculated curves correspond to the experimental data (extract from Fig. 4a), which gives the damping parameters in the range of 0.01-0.07 (the calculated deflection angles with different $F_y/F_x$ are shown in Supplementary Fig. S10). The large deflection angles that were observed in the creep regime are in sharp contrast with that observed in the current driven experiments (skyrmion Hall effect)[12], as the skyrmion Hall angles are generally suppressed in the creep regime. This behavior indicates the magnetoelastic effective field can be an additional factor besides the topological Magnus force to jointly determine the deflection angle.

By decreasing the SH wave pulse duration, we have also found the deformation of skyrmions. When the duration of the SH wave pulse is reduced to 200 ms, a few circular skyrmions deform into strip domains, as shown in Supplementary Fig. S12. One possible explanation for this phenomenon is that when the energy is insufficient to overcome pinning potentials of skyrmions, spatially inhomogeneous strain can cause the boundaries of skyrmions to move with different velocities, resulting in deformation. For longer SH wave pulses, more skyrmions are created, which leads to strong skyrmion-skyrmion repulsive interaction. As a result, a limited skyrmion motion can be identified in this case (Supplementary Fig. S12).

In summary, we have experimentally demonstrated the motion of Néel-type skyrmions driven by acoustic waves, in which the skyrmions can move along the wave propagation direction with a deflection angle with respect to the wave propagation direction, consistent with our simulations. This work provides useful insights into the control of

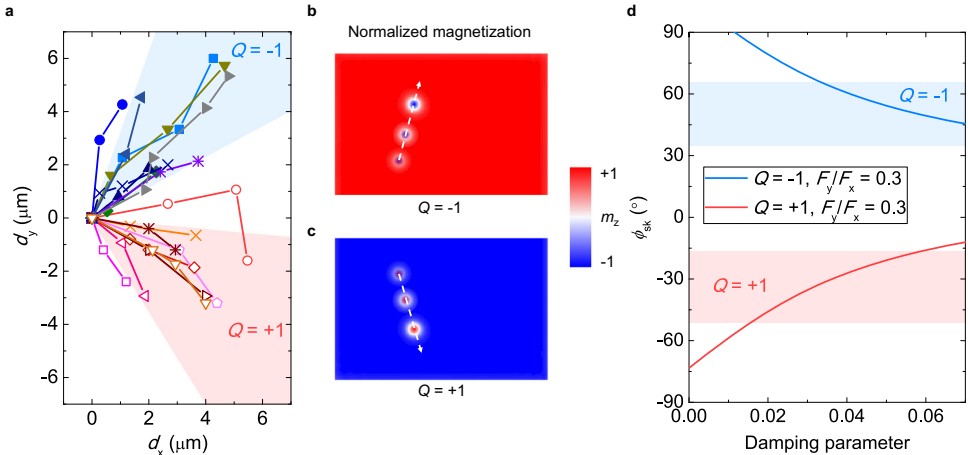

**Fig. 4 | Skyrmion motion trajectories and skyrmion deflection angles driven by SH waves. a** Motion trajectories of skyrmions driven by SH waves in experiments. **b** Simulated spatial profiles of the magnetization representing the motion of skyrmions ($Q = -1$) driven by the SH wave. **c** Simulated spatial profiles of the magnetization representing the motion of skyrmions ($Q = +1$) driven by the SH wave. **d** The experimental (blue and red regions) and numerically calculated (curves) skyrmion deflection angles ($\phi_{sk}$) versus the damping parameters, $F_y/F_x = 0.3$.

skyrmion dynamics by incorporating magnetoelastic or magneto-electric coupling. The manipulation of skyrmion dynamics by acoustic wave could potentially lead to skyrmion-based memory, logic and microwave devices without involving electric currents and with designed motion trajectory (circular motion, Supplementary Fig. S13)[47]. For a simple comparison, the efficiency of different skyrmion-driven methods is provided in Supplementary Table. S1. In particular, the efficiency of acoustic wave driven skyrmion motion can be further enhanced by implementing materials (with high magnetoelastic coupling) and devices (with high power handling). More importantly, the controlled motion trajectory with a high precision can be achieved by using high-frequency acoustic waves (small wavelength) and techniques of acoustic wave manipulation, such as phased array acoustic transducers[48]. This is comparable to acoustic tweezers for dynamic microparticle manipulation[49].

## Methods
### Sample fabrication
Synchronous two-ports SAW delay line devices were patterned on a 64°Y-cut LiNbO$_3$ substrate by using photolithography and a lift-off fabrication process. Ti (5 nm)/Pt (150 nm) electrodes were deposited on a 64°Y-cut LiNbO$_3$ substrate by using high vacuum magnetron sputtering with Ar pressure of 3 mTorr. The SH type leaky SAW is confined on the surface using IDTs consisting of heavy metal Pt electrodes on top of the LiNbO$_3$ substrate. The magnetic multilayers Ta (5 nm)/Co$_{20}$Fe$_{60}$B$_{20}$ (1 nm)/MgO (1 nm)/Ta (2 nm) were sputtered by using high vacuum magnetron sputtering with an Ar pressure of 3 mTorr.

### P-MOKE measurements with in-situ RF voltages
The skyrmions were imaged by using a polar magneto-optic Kerr effect (p-MOKE) microscope, as shown in Supplementary Fig. S14. All measurements were performed at room temperature. The transmission spectrum between two IDTs was measured using a vector network analyzer (Keysight E5080B). Radiofrequency (RF) voltage pulses supplied to IDTs were provided by an analog signal generator (Keysight N5183B) and a function/arbitrary waveform generator (Keysight 33210 A). The frequency of the RF voltage is the same as the resonance frequency of the SAW. SAW pulses are generated by an on-off keyed (OOK) RF modulation. The local heat and thermal gradient in the magnetic thin film when the RF power is applied to IDTs is evaluated by using a time-resolved thermography camera (Luxet thermo 100). See Supplementary Information Fig. S11 for details.

## Micromagnetic simulations
The micromagnetic simulation is implemented including interfacial Dzyaloshinskii-Moriya interaction, exchange interaction, magnetic anisotropy, magnetostatic, and magnetoelastic coupling contributions using MuMax3[50–52]. Considering the computational capacity limitations, the size of the simulation layer is set to be 256 nm in length, 256 nm in width, and 1 nm in height. The discretization cell size along the **x, y,** and **z** axes are 1 nm, 1 nm, and 0.5 nm, respectively. In the simulations, the diameter of skyrmions and the wavelength of the surface acoustic wave (SAW) are set to be 30 nm and 240 nm, respectively, which are significantly smaller than the corresponding values used in experiments. Nevertheless, the ratio of skyrmion diameter to the SAW wavelength remains consistent with that in experiments. The material parameters used in the simulations are as follows: an exchange constant $A_{ex} = 1 \times 10^{-11}$ J/m[20], a saturation magnetization $M_s = 5.8 \times 10^5$ A/m measured by a vibrating sample magnetometer (Supplementary Fig. S15), an interfacial Dzyaloshinskii-Moriya interaction constant $D = 3 \times 10^{-3}$ J/m$^2$, a perpendicular anisotropy constant $K_u = 7 \times 10^5$ J/m$^{3\,20}$, a Gilbert damping parameter $\alpha = 0.1$. The first order and the second order magnetoelastic coupling constants $B_1 = B_2 = -8.8 \times 10^6$ J/m$^3$ and a mass density of 8000 kg/m$^3$ are taken from the literature[44]. Elastic constants are $C_{11} = 283$ GPa, $C_{12} = 166$ GPa, and $C_{44} = 58$ GPa[44,53]. The simulations are calculated under ideal conditions without considering elastic wave attenuation. However, the wave attenuation and the pinning effect due to the impurities and defects induced disorder in actual devices will lead to skyrmion velocities driven by SAWs to be lower in reality.

## Data availability
Full data supporting the findings of this study are available within the article and its Supplementary Information. The source data generated in this study are available in the figshare repository (https://doi.org/10.6084/m9.figshare.24988164). Additional data are available from the corresponding authors upon reasonable request. Source data are provided with this paper.

## Code availability
The codes used for the micromagnetic simulations are in Supplementary Information.

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

## Acknowledgements

T.N. acknowledges funding support from the National Key R&D Program of China (Grant No. 2021YFA0716500), the National Natural Science Foundation of China (NSFC Grant Nos. 52161135103, 62131017, 52073158), and Tsinghua University Initiative Scientific Research Program. W.J. acknowledges funding support from the Basic Science Center Project of NSFC (Grant No. 51788104), the NSFC Distinguished Young Scholars (Grant No.12225409), and the National Natural Science Foundation of China (NSFC Grant Nos. 52271181, 51831005). The authors thank Truth Instruments Co., Ltd. for technical support for the Kerr microscope, and Suzhou Luxet Infrared Technology Co., Ltd. for technical support for the thermography measurement.

## Author contributions

Y.Y. and T.N. conceived the research. T.N., W.J., D.H., and D.Y. supervised the experiments. Y.Y., C.Z., D.J., and Y.J. fabricated the devices. L.Z., T.X., and W.J. deposited the thin films. Y.Y., L.Z., and Y.C. performed the measurements. Y.Y. performed the micromagnetic simulations. Y.Y. and T.N. wrote the manuscript. W.J., D.Y., D.H., P.Y., J.T., and H.W. revised the manuscript. All authors discussed the results and commented on the manuscript. T.N. directed the research.

## Competing interests

T.N. and Y.Y. have filed a patent on the technology in this paper. The remaining authors declare no competing interests.
