## [Peer Review File · Nature Communications]

Reviewers' Comments:

Reviewer #1:

Remarks to the Author:

The manuscript reports observation of skyrmion motion induced by surface acoustic waves. The general idea of moving skyrmions by SAW seems to be interesting and could potentially have applications. However, the analysis of this manuscript lacks depth, and the presentation lacks details. As a result, I do not recommend publication of this manuscript in the present form.

Questions/critics:

1. Is it clear why there is so much noise in the MOC plots, e.g., in S1c.
2. Some magnetic textures have complicated shapes and there is substantial disorder in images. Is it possible that some of magnetic textures are skyrmioniums or biskyrmions?
3. The manuscript does not offer any explanation for the observed effect while still writing some equations, such as Eqs. S3, S4. I do not see how that can explain the observed effect. Is the strain gradient necessary? As skyrmions are much smaller than the SAW wavelength from Eqs. S3 and S4 it looks like strain is enough to move skyrmions, how is it possible?
4. Equation S5 is also not useful as the paper does not calculate the force F .
5. It is not clear what kind of micromagnetic simulations have been attempted. Is it possible to provide the code?
6. The micromagnetics mentioned in the paper considers system of size 256nm x 256nm. This is much smaller than the size of skyrmion and is much, much smaller than the wavelength of SAW. how can this possibly represent your system?
7. Is it possible to provide a source of parameters used in micromagnetic simulations, how did you choose B_1 and B_2 ?

Reviewer #2:

Remarks to the Author:

The authors experimentally demonstrate the skyrmion motion driven by SAWs through shear horizontal waves instead of Rayleigh waves. These two types of SAWs are generated at the same sample area for comparison. Micromagnetic simulations confirm and support the experimental results. The skyrmion motion driven by SH waves has been systematically studied: the velocity is up to 10 μ m/s, the skyrmion Hall angles are around 49.5 and -34.2 degrees for $Q=-1$ and $Q=1$. In addition, SH waves are also more effectively than Rayleigh waves for skyrmion generation. Overall, this work shows novel experimental results, which is important for the study of skyrmion dynamics coupling with SAWs and further perspectives of low-power spintronic devices. However, before it is suitable for publication, several issues need to be addressed.

- 1) The Rayleigh waves and SH waves are generated at the same area on the 64 $^\circ$ Y-cut LiNbO₃ substrate, and the Rayleigh waves can't effectively generate skyrmions and drive the skyrmion motion. 64 $^\circ$ Y-cut LiNbO₃ substrate is usually utilized to generate shear horizontal waves (Sensors 22, 820 (2022)), is there any proof that the SAWs propagating along y direction are Rayleigh wave? Additionally, in Fig. 1i, the resonant peak for SH waves is not as obvious as Rayleigh waves, what is the reason for that?
- 2) The dominant vertical displacement of the Rayleigh waves is considered to be the main reason for the less efficiency in skyrmion motion. While in the previous works, they claimed that the longitudinal strain is dominant over other components in the thin films (Nat. Nanotechnol. 15, 361–366 (2020); Appl. Phys. Lett. 112, 112404 (2018)). The authors need to explain why the vertical displacement is mainly considered in the Rayleigh waves rather than other components in this work.

- 3) In this work, propagating Rayleigh waves and shear horizontal waves are used for experiments, and also the displacement wave form used in the simulation. However, the description of "antinode" is used in the last paragraph of Page 1. The authors need to clarify whether the standing waves are considered in simulations, which could not properly explain the experimental results.
- 4) In Fig. 2e, an abrupt increase can be seen at the RF power of 19 and 20 dBm using SH waves, what is the physical reason?
- 5) The thermal gradient generated by IDTs is lack of discussion, which has been reported in C. Chen's work (*Adv. Electron. Mater.* 8, 2200593 (2022)). The authors need to add the thermal influence on the skyrmion creation and motion.
- 6) For the skyrmion Hall angles, is there an analytical model on magnetoelastic contribution?
- 7) How to understand the skyrmion deformation after 200ms SH waves?
- 8) In Fig. S2b, the x,y,z axis does not meet the right hand spiral rule.

Response to Reviewers

We appreciate the Reviewers' interest in our work and their comments to help improve the manuscript. Both reviewers acknowledged the novelty of our experimental results while made some suggestions about different aspects. We have addressed all of their comments in the revised manuscript.

We have carefully considered all of the Reviewers' comments and modified the manuscript accordingly. Note that a revised manuscript that contains editing markups is also submitted. Below are our responses in a point-by-point manner.

REVIEWER 1 COMMENTS:

The manuscript reports observation of skyrmion motion induced by surface acoustic waves. The general idea of moving skyrmions by SAW seems to be interesting and could potentially have applications. However, the analysis of this manuscript lacks depth, and the presentation lacks details. As a result, I do not recommend publication of this manuscript in the present form.

Reply: We thank this reviewer for noting that our manuscript is interesting and could have potential applications. We also acknowledge this reviewer for her/his critical comments. Following which we have revised our manuscript accordingly. We believe the in-depth discussion, and clarity of presentations of this revised manuscript have largely enhanced. We are hoping this reviewer could now support the publication of our manuscript.

1. Is it clear why there is so much noise in the MOC plots, e.g., in S1c.

Reply: The MOKE plots show large noise for three reasons. Firstly, the substrate is transparent which limits the amplitude of reflected light. Secondly, the MOKE imaging system used in this work is reaching the limitation of the resolution and contrast of the MOKE system. The spatial resolution of our commercial MOKE system is 600 nm which is comparable to the average size of the skyrmions (1 μm) in this work. Secondly, during our SAW induced skyrmions experiments, the data quality was also influenced by the PCB mounting board for coplanar waveguides and RF cables.

2. Some magnetic textures have complicated shapes and there is substantial disorder in images. Is it possible that some of magnetic textures are skyrmioniums or biskyrmions?

Reply: We thank the reviewer for this interesting question. It is possible that some skyrmioniums would be created. However, skyrmioniums are not stable and will annihilate in our Ta/CoFeB/MgO/Ta magnetic system. The Néel-type skyrmions are stabilized by the interface Dzyaloshinskii-Moriya interaction (DMI) in non-centrosymmetric Ta/CoFeB/MgO/Ta. We have measured the interfacial DMI constant

that is about $D = 0.17 \text{ mJ/m}^2$ of the Ta/CoFeB/MgO/Ta by using the Brillouin light spectroscopy (BLS) method, as shown in Fig. R1. A biskyrmion consisting of two stable skyrmions with opposite magnetic helicities cannot be stabilized by the interface DMI. Biskyrmions usually exist in centrosymmetric materials. Therefore, the biskyrmions with the topological charge of $Q=\pm 2$ could not exist in Ta/CoFeB/MgO/Ta magnetic system. Further investigation of the disordered magnetic texture induced by acoustic waves can be imaged by XMCD-PEEM [Juge, R., et al. Skyrmions in synthetic antiferromagnets and their nucleation via electrical current and ultra-fast laser illumination. Nat Commun. 13, 4807 (2022).] or Lorentz TEM [Pollard, S., et al. Observation of stable Néel skyrmions in cobalt/palladium multilayers with Lorentz transmission electron microscopy. Nat Commun. 8, 14761 (2017).] with higher spatial resolutions. However, this is beyond the scope of this paper.

Fig. R1 | The frequency difference between the Stokes and anti-Stokes peaks versus the wavevector (k) measured by the BLS method

3. The manuscript does not offer any explanation for the observed effect while still writing some equations, such as Eqs. S3, S4. I do not see how that can explain the observed effect. Is the strain gradient necessary? As skyrmions are much smaller than the SAW wavelength from Eqs. S3 and S4 it looks like strain is enough to move skyrmions, how is it possible?

Reply: We have analyzed the underlying reasons for the SAW-driven skyrmion motion from the perspective of energy in the paragraph 3 of the main text. The total energy density of the skyrmion illustrates an asymmetric distribution under an SH wave which causes the skyrmion to move towards the lower energy density direction. As shown in Fig. 1d and g, the total energy density distribution of skyrmions under the excitation of Rayleigh wave is symmetric while it is asymmetric under the excitation of SH wave. On the other hand, the skyrmion motion can be explained by incorporating the magnetoelastic effective field that is produced by strain gradients, as suggested by Eqs. S3, S4. The strains generated by SAWs are inhomogeneous in space with strain gradients. Based on a Thiele model with the magnetoelastic force according to the references [24], [44], the forces induced by a SAW on the skyrmion in a magnetic film with an isotropy magnetostriction coefficient along the x , y and z axes are defined by

$$F_x \propto \Delta^2 \lambda_s h \left[C_{11} \frac{\partial^2 u_x}{\partial x^2} + C_{44} \left(\frac{\partial^2 u_x}{\partial y^2} + \frac{\partial^2 u_x}{\partial z^2} \right) + (C_{12} + C_{44}) \left(\frac{\partial^2 u_y}{\partial x \partial y} + \frac{\partial^2 u_z}{\partial x \partial z} \right) \right]$$

$$F_y \propto \Delta^2 \lambda_s h \left[C_{11} \frac{\partial^2 u_y}{\partial y^2} + C_{44} \left(\frac{\partial^2 u_y}{\partial x^2} + \frac{\partial^2 u_y}{\partial z^2} \right) + (C_{12} + C_{44}) \left(\frac{\partial^2 u_x}{\partial x \partial y} + \frac{\partial^2 u_z}{\partial z \partial y} \right) \right]$$

$$F_z \propto \Delta^2 \lambda_s h \left[C_{11} \frac{\partial^2 u_z}{\partial z^2} + C_{44} \left(\frac{\partial^2 u_z}{\partial x^2} + \frac{\partial^2 u_z}{\partial y^2} \right) + (C_{12} + C_{44}) \left(\frac{\partial^2 u_x}{\partial x \partial z} + \frac{\partial^2 u_y}{\partial z \partial y} \right) \right],$$

where Δ is the width of the skyrmion wall, λ_s is the magnetostriction coefficient, h is the thickness of the magnetic film, C_{ij} is the elastic constant component, u is the displacement of a SAW. The force is proportional to the strain gradient.

[24] Yanes, R., Garcia-Sanchez, F., Luis, R. F. et al. Skyrmion motion induced by voltage-controlled in-plane strain gradients. *Appl. Phys. Lett.* 115, 132401 (2019).

[44] Vanderveken, F., Mulkers, J., Leliaert, J. et al. Confined magnetoelastic waves in thin waveguides. *Phys. Rev. B* 103, 054439 (2021).

For clarity, these contents are provided in SUPPLEMENTARY INFORMATION page 7, line 20 and highlighted in yellow,

“The forces induced by an SH wave on the skyrmion in a magnetic thin film with an isotropy magnetostriction coefficient are defined by Error! Reference source not found. Error! Reference source not found.”

$$F_x \propto \Delta^2 \lambda_s h \left[C_{11} \frac{\partial^2 u_x}{\partial x^2} + C_{44} \left(\frac{\partial^2 u_x}{\partial y^2} + \frac{\partial^2 u_x}{\partial z^2} \right) + (C_{12} + C_{44}) \left(\frac{\partial^2 u_y}{\partial x \partial y} + \frac{\partial^2 u_z}{\partial x \partial z} \right) \right] \quad (S4)$$

$$F_y \propto \Delta^2 \lambda_s h \left[C_{11} \frac{\partial^2 u_y}{\partial y^2} + C_{44} \left(\frac{\partial^2 u_y}{\partial x^2} + \frac{\partial^2 u_y}{\partial z^2} \right) + (C_{12} + C_{44}) \left(\frac{\partial^2 u_x}{\partial x \partial y} + \frac{\partial^2 u_z}{\partial z \partial y} \right) \right] \quad (S5)$$

where Δ is the width of the skyrmion wall, λ_s is the magnetostriction coefficient, h is the thickness of the magnetic film, C_{ij} is the elastic constant component, u is the displacement of a SAW. F_x and F_y are proportional to the strain gradients and the magnetostriction coefficient.”

4. Equation S5 is also not useful as the paper does not calculate the force F.

Reply: Eq. S5 has been used to derive the SAW-driven skyrmion velocity (V_x and V_y) and the SAW-driven skyrmion deflection angle according to $\arctan(V_y/V_x)$. The skyrmion deflection angle is related to the ratio of the effective magnetoelastic forces along the y and x axes F_y/F_x . The force F is dependent on the strain gradient and the magnetostriction coefficient based on a Thiele model with the magnetoelastic force according to the references [24], [44].

For clarity, these contents are provided in SUPPLEMENTARY INFORMATION page 7, line 31 and highlighted in yellow,

“According to Eq. S3, the skyrmion velocity is defined by”

“From Eq. S6, the skyrmion deflection angle is defined by”

5. It is not clear what kind of micromagnetic simulations have been attempted. Is it possible to provide the code?

Reply: The details of micromagnetic simulations are described in Methods. The micromagnetic simulation is implemented using MuMax3, including interfacial Dzyaloshinskii-Moriya interaction, exchange interaction, magnetic anisotropy, magnetostatic, and magnetoelastic coupling contributions. The part of the code has been added in SUPPLEMENTARY INFORMATION page 3, line 4 and highlighted in yellow,

“The part of the code is as follows

```
d := 30
setgridsize(256, 256, 2)
Setcellsize(1e-9, 1e-9, 0.5e-9)
Msat = 580e3
Aex = 18e-12
alpha = 0.2
Dind = 3e-3
anisu = vector(0, 0, -1)
Ku1 = 6e5
Ku2 = 1.5e5
C11 = 283e9
C12 = 166e9
C44 = 58e9
rho = 8e3
B1 = -8.8e6
B2 = -8.8e6
m = Uniform(0, 0, -1)
m.setInShape(circle(d*1e-9).transl(-100e-9, 0, 0), NeelSkyrmion(-1, 1).scale(1,1,
1).transl(-100e-9, 0, 0))
relax()
for i:=0; i<256; i++){
defregion(i,xrange(-127e-9+i*1e-9,-126e-9+i*1e-9))
frozenDispLoc.SetRegion(i,1)
frozenDispVal.SetRegion(i,vector(0,1e-9*sin(2*pi/0.24e-6*i*1e-9-2*pi*486e6*dt),0))
}
SetSolver(9)
fixdt = 1e-14
autosave(m, 1e-9)
run(60e-9)
”
```

6. The micromagnetics mentioned in the paper considers system of size 256nm x 256nm. This is much smaller than the size of skyrmion and is much, much smaller than the wavelength of SAW. how can this possibly represent your system?

Reply: Due to the limitation of the computational capacity, the size of the simulation system is typically set in nanoscale. To accommodate this, the size of skyrmion and SAW wavelength in simulations are scaled down in equal proportions according to the size of skyrmion and SAW wavelength in experiments. In particular, the simulated skyrmion diameter is 30 nm, and the wavelength is 8 times as large as the skyrmion diameter in simulations. In experiments, the wavelength (8 μm) is also 8 times as large as the skyrmion diameter (1 μm). The simulation results are in a reasonable agreement with the micrometre-scale skyrmions in experiments. In the previous work of acoustic wave creation of skyrmions [37], the simulation model was also based on nanoscale skyrmions and acoustic waves with much shorter wavelength than that used in experiments.

[37] Yokouchi, T., Sugimoto, S., Rana, B. et al. Creation of magnetic skyrmion by surface acoustic waves. Nat. Nanotechnol. 15, 361-366 (2020).

For clarity, the contents are provided in page 6, line 41 right column and highlighted in yellow,

“Considering the computational capacity limitations, the size of the simulation layer is set to be 256 nm in length, 256 nm in width, and 1 nm in height.”

“In the simulations, the diameter of skyrmions and the wavelength of the surface acoustic waves (SAW) are set to be 30 nm and 240 nm, respectively, which are significantly smaller than the corresponding values used in experiments. Nevertheless, the ratio of skyrmion diameter to the SAW wavelength remains consistent with that in experiments.”

7. Is it possible to provide a source of parameters used in micromagnetic simulations, how did you choose B_1 and B_2 ?

Reply: The source of parameters used in micromagnetic simulations are shown in Methods. The magnetoelastic coupling constants B_1 and B_2 are taken from the reference [44].

[44] Vanderveken, F., Mulkers, J., Leliaert, J. et al. Confined magnetoelastic waves in thin waveguides. Phys. Rev. B 103, 054439 (2021).

For clarity, the contents are modified in page 6, line 50 right column and highlighted in yellow,

“The first order and the second order magnetoelastic coupling constants are taken from the literature⁴⁴ $B_1 = B_2 = -8.8 \times 10^6 \text{ J/m}^3$.”

REVIEWER 2 COMMENTS:

The authors experimentally demonstrate the skyrmion motion driven by SAWs through shear horizontal waves instead of Rayleigh waves. These two types of SAWs are generated at the same sample area for comparison. Micromagnetic simulations confirm and support the experimental results. The skyrmion motion driven by SH waves has been systematically studied: the velocity is up to $10\mu\text{m/s}$, the skyrmion Hall angles are around 49.5 and -34.2 degrees for $Q=-1$ and $Q=1$. In addition, SH waves are also more effectively than Rayleigh waves for skyrmion generation. Overall, this work shows novel experimental results, which is important for the study of skyrmion dynamics coupling with SAWs and further perspectives of low-power spintronic devices. However, before it is suitable for publication, several issues need to be addressed.

Reply: We appreciate this reviewer for identifying our work as novel experimental results and important for future application of skyrmionics. Stimulated by her/his comments, we have made necessary changes, which can now be found in the revised manuscript and supplementary materials.

1. The Rayleigh waves and SH waves are generated at the same area on the 64°Y -cut LiNbO_3 substrate, and the Rayleigh waves can't effectively generate skyrmions and drive the skyrmion motion. 64°Y -cut LiNbO_3 substrate is usually utilized to generate shear horizontal waves (Sensors 22, 820 (2022)), is there any proof that the SAWs propagating along y direction are Rayleigh wave? Additionally, in Fig. 1i, the resonant peak for SH waves is not as obvious as Rayleigh waves, what is the reason for that?

Reply: The mode of vibration for SAW was simulated by using a finite element method (FEM) as shown in Supplementary Fig. S2. Different modes of vibration (Rayleigh wave or SH wave) can be distinguished by analyzing the particle displacement. When the SAW propagation direction is along the y axis of the 64°Y -cut LiNbO_3 the SAW mode particle displacement consists of the vertical displacement component and the longitudinal displacement component, which indicates a Rayleigh wave mode, as shown in Supplementary Fig. S2(b). The simulated resonance frequency of the Rayleigh wave mode is also consistent with that measured in experiments.

The reason for the lower resonant peak of the SH wave than that of the Rayleigh wave is that the propagation attenuation of the SH wave mode on the 64°Y -cut LiNbO_3 is larger than that of the Rayleigh wave mode.

The reference [4] has been added in SUPPLEMENTARY INFORMATION.

For clarity, the contents are modified in SUPPLEMENTARY INFORMATION page 4, line 4 and highlighted in yellow,

“The mode of vibration for SAW was simulated by using a finite element method (FEM), as shown in Supplementary Fig. S2a, b. Different modes of vibration (Rayleigh

wave or SH wave) can be distinguished by analyzing the particle displacement. When the SAW propagation direction is along the y axis of the 64°Y -cut LiNbO_3 the SAW mode particle displacement consists of the vertical displacement component and the longitudinal displacement component, which indicates a Rayleigh wave mode, as shown in Supplementary Fig. S2b. The simulated resonance frequency of the Rayleigh wave mode is also consistent with that measured in experiments.”

[4] Mandal, D., Banerjee, S. Surface Acoustic Wave (SAW) Sensors: Physics, Materials, and Applications. Sensors, 22, 820 (2022).

Contents are modified in page 2, line 16 right column and highlighted in yellow,

“The propagation attenuation of the SH wave mode in the 64°Y -cut LiNbO_3 crystal is larger than that of the Rayleigh wave mode, resulting in a lower resonant peak for the SH wave compared to the Rayleigh wave.”

2. The dominant vertical displacement of the Rayleigh waves is considered to be the main reason for the less efficiency in skyrmion motion. While in the previous works, they claimed that the longitudinal strain is dominant over other components in the thin films (Nat. Nanotechnol. 15, 361–366 (2020); Appl. Phys. Lett. 112, 112404 (2018)). The authors need to explain why the vertical displacement is mainly considered in the Rayleigh waves rather than other components in this work.

Reply: Previous works have used the longitudinal strain component of Rayleigh waves in their analysis. The dominant displacement components of Rayleigh waves are quite different depending on different cut types of piezoelectric substrates. The cut type of LiNbO_3 in our device is different from the previous work (Nat. Nanotechnol. 15, 361–366 (2020); Appl. Phys. Lett. 112, 112404 (2018)). In our FEM simulation, we find that the vertical displacement of the Rayleigh wave is 1.5 times as large as its longitudinal displacement along x when the wave propagation direction is along the y axis of 64°Y -cut LiNbO_3 , as shown in Fig. S3. We have thus mainly considered the vertical displacement of the Rayleigh waves in the micromagnetic simulations as shown in Fig. 1b. We have also analyzed whether the longitudinal displacement along x alone or a combination of longitudinal and vertical displacement of Rayleigh wave can move skyrmions. We found that they both move skyrmions much less effectively than the particle displacement along y of the SH wave. The skyrmions driven by Rayleigh wave are usually trapped at the maximum particle displacement within one wavelength.

For clarity, the contents are modified in SUPPLEMENTARY INFORMATION page 4, line 32 and highlighted in yellow,

“The dominant displacement components of Rayleigh waves are different depending on different cut types of piezoelectric substrates. In our FEM simulation, we find that the vertical displacement of the Rayleigh wave is 1.5 times as large as its longitudinal displacement along x when the wave propagation direction is along the y axis of 64°Y -cut LiNbO_3 , as shown in Fig. S3. We have thus mainly considered the vertical displacement of the Rayleigh waves in the micromagnetic simulations as shown in Fig. 1b. We have also analyzed whether the longitudinal displacement along x alone or a

combination of longitudinal and vertical displacement of Rayleigh wave can move skyrmions. We found that they both move skyrmions much less effectively than the particle displacement along y of the SH wave. The skyrmions driven by Rayleigh wave are usually trapped at the maximum particle displacement within one wavelength.

Supplementary Fig. S3 | The simulated particle displacement components of the Rayleigh wave when the wave propagation direction is along the y axis of 64° Y-cut LiNbO_3 . The longitudinal displacement component (L), the shear horizontal displacement component (SH), the vertical displacement component (V).

3. In this work, propagating Rayleigh waves and shear horizontal waves are used for experiments, and also the displacement wave form used in the simulation. However, the description of “antinode” is used in the last paragraph of Page 1. The authors need to clarify whether the standing waves are considered in simulations, which could not properly explain the experimental results.

Reply: We apologize for the unclear description. The propagating wave is considered in simulations. We mean that the skyrmions would be trapped at the maximum particle displacement of the Rayleigh wave. The sentence has been modified.

For clarity, the contents are modified in page 1, line 50 right column and highlighted in yellow,

“We observe that the Rayleigh wave is capable of moving skyrmions only for a distance of less than half an acoustic wavelength, after which the skyrmions become trapped at the maximum particle displacement of the Rayleigh wave.”

4. In Fig. 2e, an abrupt increase can be seen at the RF power of 19 and 20 dBm using SH waves, what is the physical reason?

Reply: There are pinning potentials for the generation of skyrmions which are induced by impurities and defects in the magnetic film. The spacing of these impurities and defects is often smaller than the magnetic skyrmion size. Therefore, the power of the SH wave has to be higher than a threshold to create more skyrmions by overcoming the energy barrier. Above the energy barrier, more skyrmions can be created which shows an abrupt increase in Fig. 2e. While for the Rayleigh wave, we do not observe any

threshold effect since the in-plane magnetoelastic energy generated by the Rayleigh wave is always lower than the energy barrier.

5. The thermal gradient generated by IDTs is lack of discussion, which has been reported in C. Chen's work (Adv. Electron. Mater. 8, 2200593 (2022)). The authors need to add the thermal influence on the skyrmion creation and motion.

Reply: We thank the reviewer for this critical question. The skyrmions can indeed be generated and moved by a substantial thermal gradient [20]. We have now carefully studied the amplitude of thermal gradient in our sample generated by IDTs using a time-resolved thermography camera (SUPPLEMENTARY INFORMATION [9]), as shown in Supplementary Fig. S8. In our sample, the distance from the edge of the magnetic film to the edge of the interdigital transducer (IDT) is 62 μm . We have taken temperature images after the excitation of SAW with a pulse duration of 300 ms and a RF power of 26 dBm at the room temperature, in which these experimental parameters are what we normally use for the generation and movement of skyrmions. We found that the temperature rise in the magnetic film is less than 1 $^{\circ}\text{C}$, which is consistent with the reports in the reference [37]. Such temperature rise would have little influence on the magnetic properties of magnetic thin films. The generated thermal gradient in the magnetic film is 1.2 $\text{mK}/\mu\text{m}$, which is about 1000 times smaller than the value that can generate or move skyrmions (according to the reference [20], thermal gradients of over 1.6 $\text{K}/\mu\text{m}$ and 0.35-0.56 $\text{K}/\mu\text{m}$ are required to generate and move skyrmions). In conclusion, the thermal gradient generated by IDTs is negligible to influence the skyrmion generation and motion.

The reference [Adv. Electron. Mater. 8, 2200593 (2022)] has been added in SUPPLEMENTARY INFORMATION.

[20] Wang, Z., Guo, M., Zhou, H.A. et al. Thermal generation, manipulation and thermoelectric detection of skyrmion. Nat. Electron. 3, 672-679 (2020).

[37] Yokouchi, T., Sugimoto, S., Rana, B. et al. Creation of magnetic skyrmion by surface acoustic waves. Nat. Nanotechnol. 15, 361-366 (2020).

SUPPLEMENTARY INFORMATION [8] Chen, C., Fu, S., Han, L., et al. Energy Harvest in Ferromagnet-Embedded Surface Acoustic Wave Devices. Adv. Electron. Mater. 8, 2200593 (2022).

SUPPLEMENTARY INFORMATION [9] O. Breitenstein, W. Warta, and M. Langenkamp, Lock-in Thermography: Basics and Use for Evaluating Electronic Devices and Materials (Springer, Berlin/Heidelberg, 2010).

For clarity, the contents are modified in page 6, line 35 right column and highlighted in yellow,

“The local heat and thermal gradient in the magnetic thin film when the RF power is applied to IDTs is evaluated by using a time-resolved thermography camera (Luxet thermo 100). See Supplementary Information Fig. S8 for details.”

Contents are modified in SUPPLEMENTARY INFORMATION page 8, line 24 and highlighted in yellow,

“When the RF power is applied to IDTs, local heat can be generated in the magnetic thin film, resulting in the establishment of a thermal gradient⁸. We quantified the amplitude of the thermal gradient in the sample induced by the RF power using a time-resolved thermography camera⁹, as shown in Fig. S8. In our sample, the distance from the edge of the magnetic film to the edge of the IDT is 62 μm . Temperature images were captured after exciting SAW with a pulse duration of 300 ms and an RF power of 26 dBm at room temperature, which are the standard experimental parameters for skyrmion generation and motion. Our finding reveal that the temperature rise in the magnetic film is less than 1 $^{\circ}\text{C}$, and therefore would have minimal impact on the magnetic properties of the thin film. The generated thermal gradient in the magnetic film is measured to be 1.2 mK/ μm , indicating that the thermal gradient induced by IDTs is negligible in influencing the skyrmion generation and motion¹⁰.”

Supplementary Fig. S8 | The temperature rise image of the device at the room temperature when a 300 ms RF pulse with the power of 26 dBm are applied to the IDT. The scale bar is 100 μm .”

6. For the skyrmion Hall angles, is there an analytical model on magnetoelastic contribution?

Reply: The skyrmion motion driven by the SAW can be described by a Thiele equation. The skyrmion deflection angle driven by the SAW is defined by

$$\phi_{sk} = \arctan \left(\frac{G + \alpha D \frac{F_y}{F_x}}{\alpha D - G \frac{F_y}{F_x}} \right)$$

F_x and F_y are the effective magnetoelastic forces along the x and y axes with magnetoelastic contribution, respectively. They are dependent on the strain gradient and the magnetostriction coefficient. According to a Thiele model with the magnetoelastic force from the references [24], [44], the forces induced by an SH wave on the skyrmion in an isotropy magnetic film along the x and y axes are defined by

$$F_x \propto \Delta^2 \lambda_s h \left[C_{11} \frac{\partial^2 u_x}{\partial x^2} + C_{44} \left(\frac{\partial^2 u_x}{\partial y^2} + \frac{\partial^2 u_x}{\partial z^2} \right) + (C_{12} + C_{44}) \left(\frac{\partial^2 u_y}{\partial x \partial y} + \frac{\partial^2 u_z}{\partial x \partial z} \right) \right]$$

$$F_y \propto \Delta^2 \lambda_s h \left[C_{11} \frac{\partial^2 u_y}{\partial y^2} + C_{44} \left(\frac{\partial^2 u_y}{\partial x^2} + \frac{\partial^2 u_y}{\partial z^2} \right) + (C_{12} + C_{44}) \left(\frac{\partial^2 u_x}{\partial x \partial y} + \frac{\partial^2 u_z}{\partial z \partial y} \right) \right]$$

where Δ is the width of the skyrmion wall. λ_s is the magnetostriction coefficient. h is the ferromagnetic film thickness. C_{ij} is the elastic constant component. u_y is the displacement of an SH wave. The magnetostriction coefficient is related to the magnetoelastic coupling constant. If the magnetic film has an isotropy magnetoelastic coupling constant, the skyrmion deflection angle is obtained as

$$\phi_{sk} = \arctan \left(\frac{G + \alpha D \left[C_{11} \frac{\partial^2 u_y}{\partial y^2} + C_{44} \left(\frac{\partial^2 u_y}{\partial x^2} + \frac{\partial^2 u_y}{\partial z^2} \right) + (C_{12} + C_{44}) \left(\frac{\partial^2 u_x}{\partial x \partial y} + \frac{\partial^2 u_z}{\partial z \partial y} \right) \right]}{C_{11} \frac{\partial^2 u_x}{\partial x^2} + C_{44} \left(\frac{\partial^2 u_x}{\partial y^2} + \frac{\partial^2 u_x}{\partial z^2} \right) + (C_{12} + C_{44}) \left(\frac{\partial^2 u_y}{\partial x \partial y} + \frac{\partial^2 u_z}{\partial x \partial z} \right)} \right) \cdot \frac{C_{11} \frac{\partial^2 u_y}{\partial y^2} + C_{44} \left(\frac{\partial^2 u_y}{\partial x^2} + \frac{\partial^2 u_y}{\partial z^2} \right) + (C_{12} + C_{44}) \left(\frac{\partial^2 u_x}{\partial x \partial y} + \frac{\partial^2 u_z}{\partial z \partial y} \right)}{\alpha D - G \left[C_{11} \frac{\partial^2 u_x}{\partial x^2} + C_{44} \left(\frac{\partial^2 u_x}{\partial y^2} + \frac{\partial^2 u_x}{\partial z^2} \right) + (C_{12} + C_{44}) \left(\frac{\partial^2 u_y}{\partial x \partial y} + \frac{\partial^2 u_z}{\partial x \partial z} \right) \right]}$$

The magnetostriction coefficient is cancelled out in the skyrmion deflection angle equation. The skyrmion deflection angle has a strong correlation to the elastic constants and strain gradients. We have also tried to use different magnetoelastic coupling constants in simulations and found that the skyrmion deflection angle remained unchanged.

[24] Yanes, R., Garcia-Sanchez, F., Luis, R. F. et al. Skyrmion motion induced by voltage-controlled in-plane strain gradients. *Appl. Phys. Lett.* 115, 132401 (2019).

[44] Vanderveken, F., Mulkers, J., Leliaert, J. et al. Confined magnetoelastic waves in thin waveguides. *Phys. Rev. B* 103, 054439 (2021).

Contents are modified in SUPPLEMENTARY INFORMATION page 8, line 1 and highlighted in yellow,

“For magnetic thin films with an isotropy magnetoelastic coupling constant, the magnetostriction coefficient would be nullified in the equation for skyrmion deflection angle. In our simulations, we also tested using various magnetoelastic coupling constants but observed no change in the skyrmion deflection angle.”

7. How to understand the skyrmion deformation after 200ms SH waves?

Reply: The energy of an SH wave with 200 ms pulse duration is not enough for skyrmions to overcome the pinning potentials. Because the skyrmion diameter is around 1 μm that is one eighth of the wavelength, the strain is spatially inhomogeneous on the skyrmion. The boundaries of the skyrmion could have different velocities leading to the skyrmion deformation.

For clarity, the contents are modified in page 4, line 13 right column and highlighted in yellow,

“One possible explanation for this phenomenon is that when the energy is insufficient to overcome pinning potentials of skyrmions, spatially inhomogeneous strain can cause the boundaries of the skyrmion to move at different velocities, resulting in its deformation.”

8. In Fig. S2b, the x,y,z axis does not meet the right hand spiral rule.

Reply: The coordination system has now been modified to meet the right-hand spiral rule in Fig. S2b.

For clarity, the contents are modified in SUPPLEMENTARY INFORMATION page 4, line 12 and highlighted in yellow,

“

Supplementary Fig. S2 | b, The simulated displacement distribution of the Rayleigh wave.”

Reviewers' Comments:

Reviewer #1:

Remarks to the Author:

After reading the revised paper, I do not recommend publication of this article. Some of questions have not been answered. Furthermore, the model of magnetoelastic coupling in Appl. Phys. Lett. 115, 132401 (2019) might not be applicable to the setup in the article. There is also no shear strain present in Appl. Phys. Lett. 115, 132401 (2019). The code that authors shared does not contain any detail and is only a small part of the whole code.

With the level of details provided I have no way of determining the correctness of this paper. I suspect that the model of magnetoelastic coupling used in this article might not be applicable to CoFeB.

Reviewer #2:

Remarks to the Author:

The authors give detail explanation about the existence of Rayleigh waves in their samples and careful discussion about the influence of thermal effects on the skyrmion motion. The experimental results are clear and interesting; however, there are some minor issues should be clarified before it can be considered for publication.

1) The authors claim that "the Rayleigh wave is capable of moving skyrmions only for a distance of less than half an acoustic wavelength, after which the skyrmions become trapped at the maximum particle displacement of the Rayleigh wave". While, the velocity of SAWs propagating along piezoelectric substrates is always much larger than that of skyrmions. It does not seem to be reasonable that skyrmion could be trapped at the maximum particle displacement or other position in the wave. If so, a velocity of skyrmion motion up to thousand meter per second would be achieved. Authors need to clarify this statement and provide some details about the simulation.

2) The authors should give the SAW expression in their simulation like their previous work [J. Phys. D: Appl. Phys. 56, 084002 (2023)], an analytical model or a phenomenological model would be helpful for understanding.

3) The authors mention that "the dominant displacement components of Rayleigh waves are quite different depending on different cut types of piezoelectric substrates", a reference is necessary to support this statement.

Response to Reviewers

We appreciate the both reviewers for their valuable comments and interests in our work, which have helped us to further improve the both quality and clarity of our work.

Stimulated by their insightful suggestions, we have carefully revised our manuscript accordingly. Note that a revised manuscript that contains editing markups is also submitted. Below are our responses in a point-by-point manner.

REVIEWER 1 COMMENTS:

Reviewer comments: After reading the revised paper, I do not recommend publication of this article. Some of questions have not been answered.

Reply: We thank this reviewer for critical reading of our manuscript and for making useful suggestions. We particularly apologize for leaving an impression of “not answering some of the questions”. In fact, in our previous response letter, we have answered the reviewer’s questions point by point. On the other hand, we have recognized that part of our early responses may be insufficient in addressing some of the comments. In order to respond to this particular comment, we have substantially revised our early responses, which can now be found below. We are hoping this reviewer are now convinced by the robustness of our work and in the position of supporting the publication.

Below we enlisted our 1st round responses to this reviewer:

1. Is it clear why there is so much noise in the MOC plots, e.g., in S1c.

Reply: We thank this reviewer for this piece of comment. This is because: 1. Substrate transparency: The substrate LiNbO_3 we used in this experiment is transparent, which can limit the amplitude of reflected light. 2. Limitation of the MOKE imaging system: The spatial resolution of our commercial MOKE system is 600 nm, which is comparable to the average size of the skyrmions investigated in our study (1 μm). 3. Influence of PCB Mounting Board: During our experiments, the mechanical vibrations and electromagnetic interference introduced by the mounting board and RF cables can contribute to the noise observed in the MOKE measurements. While we made efforts to minimize these effects, we understand that they can still impact the data to some extent.

2. Some magnetic textures have complicated shapes and there is substantial disorder in images. Is it possible that some of magnetic textures are skyrmioniums or biskyrmions?

Reply: It is possible that some skyrmioniums could be created in our system. However, it is important to note that skyrmioniums are not stable and will ultimately annihilate in our Ta/CoFeB/MgO/Ta magnetic system. The stability of Néel-type skyrmions is primarily attributed to the interface Dzyaloshinskii-Moriya interaction (DMI) in non-centrosymmetric materials like Ta/CoFeB/MgO/Ta. We have measured the interfacial DMI constant that is about $D = 0.17 \text{ mJ/m}^2$ of the Ta/CoFeB/MgO/Ta by using the Brillouin light spectroscopy (BLS) method. A biskyrmion consisting of two stable skyrmions with opposite magnetic helicities cannot be stabilized by the interface DMI. Biskyrmions are typically observed in centrosymmetric materials, and therefore, it is unlikely for biskyrmions with a topological charge of $Q=\pm 2$ to exist in our Ta/CoFeB/MgO/Ta magnetic system. While further investigation of the disordered magnetic texture induced by acoustic waves can be imaged with higher spatial resolutions using techniques such as XMCD-PEEM [Juge, R., et al. Skyrmions in synthetic antiferromagnets and their nucleation via electrical current and ultra-fast laser illumination. Nat Commun. 13, 4807 (2022).] or Lorentz TEM [Pollard, S., et al. Observation of stable Néel skyrmions in cobalt/palladium multilayers with Lorentz transmission electron microscopy. Nat Commun. 8, 14761 (2017).]. However, due to the limited spatial resolution, our MOKE microscope cannot resolve the presence of skyrmionium in the present material system.

3. *The manuscript does not offer any explanation for the observed effect while still writing some equations, such as Eqs. S3, S4. I do not see how that can explain the observed effect. Is the strain gradient necessary? As skyrmions are much smaller than the SAW wavelength from Eqs. S3 and S4 it looks like strain is enough to move skyrmions, how is it possible?*

Reply: We apologize for the lack of clarity in the previous response letter. We appreciate your thorough evaluation of the equations presented in the manuscript. We will provide a more detailed explanation here.

The strain gradient is indeed important and necessary to understand the observed effect of skyrmion motion driven by surface acoustic waves (SAWs). The movement of skyrmions in response to SAWs can be explained by the effective force generated by strain gradients. Based on a Thiele model incorporating magnetoelastic force, the forces acting on a skyrmion in a magnetic film with an isotropy magnetostriction coefficient are proportional to strain gradients:

$$F_x = A_{xx}\nabla\epsilon_{xx} + A_{xy}\nabla\epsilon_{xy} + A_{xz}\nabla\epsilon_{xz} + A_{yx}\nabla\epsilon_{yx} + A_{zx}\nabla\epsilon_{zx} \quad (\text{S6})$$

$$F_y = A_{yy}\nabla\epsilon_{yy} + A_{yx}\nabla\epsilon_{yx} + A_{yz}\nabla\epsilon_{yz} + A_{xy}\nabla\epsilon_{xy} + A_{zy}\nabla\epsilon_{zy} \quad (\text{S7})$$

where A_{ii} is a coefficient depending on the spin texture of a skyrmion. A_{xx} and A_{yy} depend on the first order magnetoelastic coupling constant B_1 . A_{xy} , A_{xz} , A_{zx} , A_{yx} , A_{yz} , and

A_{zy} depend on the second order magnetoelastic coupling constant B_2 . ε_{ii} is a strain component induced by a SAW. The strains generated by SAWs are inhomogeneous in space with strain gradients. If the strain were static and constant without any strain gradients in space, the skyrmion would remain stationary.

Furthermore, the strain gradients also lead to an inhomogeneous distribution of magnetoelastic energy density in space. In the manuscript, we also analyze the reasons for the SAW-driven skyrmion motion from an energy perspective. The total energy density of the skyrmion illustrates an asymmetric distribution under an SH wave which causes the skyrmion to move towards the lower energy density direction. As shown in Fig. 1d and g, the total energy density distribution of skyrmions under the excitation of Rayleigh wave is symmetric while it is asymmetric under the excitation of SH wave.

4. Equation S5 is also not useful as the paper does not calculate the force F .

Reply: We apologize for any confusion caused by the use of Equation S5 without providing a calculation for the force F . Allow us to provide a more detailed explanation. Eq. S5 has been employed to derive the SAW-driven skyrmion velocity (V_x and V_y) and the SAW-driven skyrmion deflection angle, calculated as $\arctan(V_y/V_x)$. The skyrmion deflection angle is related to the ratio of the effective magnetoelastic forces along the y and x axes (F_y/F_x). The force F is dependent on the strain gradient and the magnetostriction coefficient as described by a Thiele model incorporating magnetoelastic force.

$$F_x = A_{xx}\nabla\varepsilon_{xx} + A_{xy}\nabla\varepsilon_{xy} + A_{xz}\nabla\varepsilon_{xz} + A_{yx}\nabla\varepsilon_{yx} + A_{zx}\nabla\varepsilon_{zx} \quad (S6)$$

$$F_y = A_{yy}\nabla\varepsilon_{yy} + A_{yx}\nabla\varepsilon_{yx} + A_{yz}\nabla\varepsilon_{yz} + A_{xy}\nabla\varepsilon_{xy} + A_{zy}\nabla\varepsilon_{zy} \quad (S7)$$

where A_{ii} is a coefficient depending on the spin texture of a skyrmion. A_{xx} and A_{yy} depend on the first order magnetoelastic coupling constant B_1 . A_{xy} , A_{xz} , A_{zx} , A_{yx} , A_{yz} , and A_{zy} depend on the second order magnetoelastic coupling constant B_2 . E_{ii} is a strain component induced by a SAW. We acknowledge that the force F plays a crucial role in the analysis; however, performing an analytical calculation of F is challenging due to unknown material constants and the complexity of the boundary conditions for thin films and the piezoelectric substrate. Due to these complexities and limitations, we estimate the ratio F_y/F_x rather than providing an explicit calculation. This estimation allows us to analyze the SAW-driven skyrmion behavior and make meaningful interpretations based on the available information.

5. It is not clear what kind of micromagnetic simulations have been attempted. Is it possible to provide the code?

Reply: We appreciate your question regarding the specifics of the micromagnetic simulations conducted in our study. The details of these simulations are described in

Methods. The micromagnetic simulation were implemented using MuMax3, which incorporates energies including interfacial Dzyaloshinskii-Moriya interaction, exchange interaction, magnetic anisotropy, magnetostatic, and magnetoelastic coupling. The complete code for the simulation has been added in SUPPLEMENTARY INFORMATION page 3, line 4 and highlighted in yellow.

6. *The micromagnetics mentioned in the paper considers system of size 256nm x 256nm. This is much smaller than the size of skyrmion and is much, much smaller than the wavelength of SAW. how can this possibly represent your system?*

Reply: The choice of the simulation system size is influenced by the limitations of computational capacity. Typically, due to computational constraints, the size of the simulated system is set in the nanoscale. To address this limitation, the size of skyrmion and SAW wavelength in simulations are scaled down in equal proportions according to the size of skyrmion and SAW wavelength in experiments. Specifically, in our simulation, the diameter of the skyrmion is 30 nm, and the wavelength is set to be 8 times larger than the skyrmion diameter. In the experiments, the wavelength (8 μm) is also 8 times larger than the skyrmion diameter (1 μm). We found that the simulation results are in a reasonable agreement with the experimental observations of micrometer-scale skyrmions. It is important to note that in the previous work of acoustic wave creation of skyrmions [37], the simulation model also employed nanoscale skyrmions and acoustic waves with significantly shorter wavelengths than those used in early experiments.

[37] Yokouchi, T., Sugimoto, S., Rana, B. et al. Creation of magnetic skyrmion by surface acoustic waves. *Nat. Nanotechnol.* 15, 361-366 (2020).

7. *Is it possible to provide a source of parameters used in micromagnetic simulations, how did you choose B_1 and B_2 ?*

Reply: The source of parameters used in micromagnetic simulations are shown in Methods. The magnetoelastic coupling constants B_1 and B_2 are taken from the reference [44].

[44] Vanderveken, F., Mulkers, J., Leliaert, J. et al. Confined magnetoelastic waves in thin waveguides. *Phys. Rev. B* 103, 054439 (2021).

Below are our responses to the 2nd round comments:

Reviewer comments: Furthermore, the model of magnetoelastic coupling in Appl. Phys. Lett. 115, 132401 (2019) might not be applicable to the setup in the article. There is also no shear strain present in Appl. Phys. Lett. 115, 132401 (2019).

Reply: We appreciate this valuable comment. Following which, we have now carefully considered its consequence and make proper changes in this revised manuscript. The

model of magnetoelastic coupling in the reference [24] is a simplified model that consider only static strain ε_{xx} along the x -axis. In our study, we have extended this model by incorporating the magnetoelastic coupling of SAWs. While shear strain is not explicitly mentioned in the reference [24], we have expanded the model to include shear strain gradients as part of the effective forces (F_x, F_y) in the Thiele model. Furthermore, the reference [38] presents the effective force F_x of the Thiele model, considering the magnetoelastic coupling induced by SAWs. The effective force in reference [38] is proportional to strain gradients, including shear gradients. In our extended model, we have derived equivalent expressions for the effective forces, in consistency with the formulations in reference [38]. In particular, the effective forces read as follows:

$$F_x = A_{xx}\nabla\varepsilon_{xx} + A_{yy}\nabla\varepsilon_{yy} + A_{xz}\nabla\varepsilon_{xz} + A_{yx}\nabla\varepsilon_{yx} + A_{zx}\nabla\varepsilon_{zx} \quad (S6)$$

$$F_y = A_{yy}\nabla\varepsilon_{yy} + A_{yx}\nabla\varepsilon_{yx} + A_{yz}\nabla\varepsilon_{yz} + A_{xy}\nabla\varepsilon_{xy} + A_{zy}\nabla\varepsilon_{zy} \quad (S7)$$

where A_{ii} is a coefficient depending on the spin texture of a skyrmion. A_{xx} and A_{yy} depend on the first order magnetoelastic coupling constant B_1 . A_{xy} , A_{xz} , A_{zx} , A_{yx} , A_{yz} , and A_{zy} depend on the second order magnetoelastic coupling constant B_2 . ε_{ii} is a strain component induced by a SAW. This analytical model allows us to compute the skyrmion deflection angle driven by SAWs

$$\phi_{sk} = \arctan \left(\frac{G + \alpha D \frac{F_y}{F_x}}{\alpha D - G \frac{F_y}{F_x}} \right) \quad (S9)$$

The skyrmion deflection angle is related to the ratio F_y/F_x . It is important to note that the calculation of the skyrmion deflection angle does not require determining the precise values of F_x and F_y . And calculating the exact values of F_x and F_y is challenging due to complex boundary conditions for thin films and the piezoelectric substrate. We estimate the ratio F_y/F_x in our model. The validity of our extended model is being well justified by performing micromagnetic simulation, which can be found from the main text.

[24] Yanes, R., Garcia-Sanchez, F., Luis, R. F. et al. Skyrmion motion induced by voltage-controlled in-plane strain gradients. *Appl. Phys. Lett.* 115, 132401 (2019).

[38] Nepal, R., GÜNGÖRDÜ, U., Kovalev, A. A. Magnetic skyrmion bubble motion driven by surface acoustic waves. *Appl. Phys. Lett.* 112, 112404 (2018).

To deliver a clear message, these contents are properly modified and presented in SUPPLEMENTARY INFORMATION page 8, line 41 and highlighted in yellow,

“The effective forces induced by an SH wave on the skyrmion in a magnetic thin film with an isotropy magnetostriction coefficient are defined by⁷⁻¹⁰

$$F_x = A_{xx}\nabla\varepsilon_{xx} + A_{yy}\nabla\varepsilon_{yy} + A_{xz}\nabla\varepsilon_{xz} + A_{yx}\nabla\varepsilon_{yx} + A_{zx}\nabla\varepsilon_{zx} \quad (S6)$$

$$F_y = A_{yy}\nabla\epsilon_{yy} + A_{yx}\nabla\epsilon_{yx} + A_{yz}\nabla\epsilon_{yz} + A_{xy}\nabla\epsilon_{xy} + A_{zy}\nabla\epsilon_{zy} \quad (S7)$$

where A_{ii} is a coefficient depending on the spin texture of a skyrmion^{7,10}. A_{xx} and A_{yy} depend on the first order magnetoelastic coupling constant B_1 . A_{xy} , A_{xz} , A_{zx} , A_{yx} , A_{yz} , and A_{zy} depend on the second order magnetoelastic coupling constant B_2 ^{7,10}. ϵ_{ii} is a strain component induced by a SAW.”

Reviewer comments: The code that authors shared does not contain any detail and is only a small part of the whole code.

Reply: We apologize for not presenting sufficient details of our simulation package. The micromagnetic simulation is implemented using an extension of MuMax3, which includes the magnetoelastodynamics and magnetoelastic coupling solver [44], [51]. These references [44], [51] have been cited in both the original main text and SUPPLEMENTARY INFORMATION. We have also included the code segments that define the SH wave, the Rayleigh wave, along with corresponding code comments to enhance code readability and understanding. The complete code and code comments have been included in SUPPLEMENTARY INFORMATION page 3, line 4 and highlighted in yellow,

```

“The whole code and code comments for an SH wave are as follows
d := 30
Setgridsize(256, 256, 2)
Setcellsize(1e-9, 1e-9, 0.5e-9) // Define the geometry and meshes
Msat = 580e3 // Define a saturation magnetization
Aex = 10e-12 // Define an exchange constant
alpha = 0.1 // Define a damping parameter
Dind = 3e-3 // Define an interfacial Dzyaloshinskii-Moriya
constant
anis = vector(0, 0, -1) // Define the uniaxial anisotropy direction
Ku1 = 7e5 // Define the first order uniaxial anisotropy constant
Ku2 = 1.5e5 // Define the second order uniaxial anisotropy constant
C11 = 283e9 // Define elastic constants
C12 = 166e9
C44 = 58e9
rho = 8e3
B1 = -8.8e6 // Define the first order magnetoelastic coupling constant
B2 = -8.8e6 // Define the second order magnetoelastic coupling constant
Temp = 0 // Temperature
m = Uniform(0, 0, -1) // Initial magnetization state
m.setInShape(circle(d*1e-9).transl(-50e-9, 0, 0), NeelSkyrmion(-1, 1).scale(1,1,
1).transl(-50e-9, 0, 0))
// define the skyrmion size, type and position

```

```

relax() // Minimize the total energy
for i:=0; i<256; i++){
defregion(i,xrange(-127e-9+i*1e-9,-126e-9+i*1e-9))
frozenDispLoc.SetRegion(i,1)
frozenDispVal.SetRegion(i,vector(0,1e-9*sin(2*pi/0.24e-6*i*1e-9-2*pi*486e6*t),0))
// Define an SH wave
}
SetSolver(9) // Solver with magnetoelastic interaction
fixdt = 1e-14 // Set a fixed time step
autosave(m, 1e-9) // Magnetization output
autosave(Edens_mel, 1e-9) // Magnetoelastic energy density output
autosave(Edens_total, 1e-9) // Total energy density output
run(400e-9) // Running

```

The whole code and code comments for a Rayleigh wave are as follows

```

d := 30
Setgridsize(256, 256, 2)
Setcellsize(1e-9, 1e-9, 0.5e-9) // Define the geometry and meshes
Msat = 580e3 // Define a saturation magnetization
Aex = 10e-12 // Define an exchange constant
alpha = 0.1 // Define a damping parameter
Dind = 3e-3 // Define an interfacial Dzyaloshinskii-Moriya
constant
anisu = vector(0, 0, -1) // Define the uniaxial anisotropy direction
Ku1 = 7e5 // Define the first order uniaxial anisotropy constant
Ku2 = 1.5e5 // Define the second order uniaxial anisotropy constant
C11 = 283e9 // Define elastic constants
C12 = 166e9
C44 = 58e9
rho = 8e3
B1 = -8.8e6 // Define the first order magnetoelastic coupling constant
B2 = -8.8e6 // Define the second order magnetoelastic coupling constant
Temp = 0 // Temperature
m = Uniform(0, 0, -1) // Initial magnetization state
m.setInShape(circle(d*1e-9).transl(-50e-9, 0, 0), NeelSkyrmion(-1, 1).scale(1,1,
1).transl(-50e-9, 0, 0))
// define the skyrmion size, type and position
relax() // Minimize the total energy
for i:=0; i<256; i++){
defregion(i,xrange(-127e-9+i*1e-9,-126e-9+i*1e-9))
frozenDispLoc.SetRegion(i,1)

```

```

frozenDispVal.SetRegion(i,vector(-1e-9*2/3*cos(2*pi/0.24e-6*i*1e-9-
2*pi*451e6*t),0,1e-9*sin(2*pi/0.24e-6*i*1e-9-2*pi*451e6*t)))
// Define a Rayleigh wave
}
SetSolver(9) // Solver with magnetoelastic interaction
fixdt = 1e-14 // Set a fixed time step
autosave(m, 1e-9) // Magnetization output
autosave(Edens_mel, 1e-9) // Magnetoelastic energy density output
autosave(Edens_total, 1e-9) // Total energy density output
run(400e-9) // Running
”

```

[44] Vanderveken, F., Mulkers, J., Leliaert, J. et al. Confined magnetoelastic waves in thin waveguides. Phys. Rev. B 103, 054439 (2021).

[51] Vanderveken, F., Mulkers, J., Leliaert, J. et al. Finite difference magnetoelastic simulator. Open Research Europe 1, 1-23 (2021).

Reviewer comments: With the level of details provided I have no way of determining the correctness of this paper. I suspect that the model of magnetoelastic coupling used in this article might not be applicable to CoFeB.

Reply: We appreciate your concern regarding the applicability of the magnetoelastic coupling model used in our article. We would like to clarify that our manuscript focuses predominantly on the experimental aspect, with the model serving as a tool to explain the observed results. The model of magnetoelastic coupling we employed is a widely accepted and applicable model for ferromagnetic films. Of course, the concern from this reviewer is correct, the specific material parameters used in the model may vary depending on the material under investigation.

In our study, the model of magnetoelastic coupling is adapted from the reference [24], [44]. The reference [24] explicitly studied CoFeB. The saturation magnetization is measured by a vibrating sample magnetometer, as shown in Supplementary Fig. S13. The material parameters, including the saturation magnetization, exchange constant, perpendicular anisotropy constant, mass density, first-order and second-order magnetoelastic coupling constants, and elastic constants, are determined based on experimental measurements and references [20], [44], and [52]. Our analytical model has used the same material parameters as in the micromagnetic simulation. The reference [44] also considers CoFeB in the simulation using a MuMax3 extension with the magnetoelastodynamics and magnetoelastic coupling solver, supporting its applicability to our system.

Note that our focus is to demonstrate experimentally the SAWs driven skyrmions motion, and the model serves as a complementary tool for interpretation. We have

directly visualized the skyrmion motion trajectories and provided strong experimental evidences that the skyrmions deflect with opposite angles depending on their topological charges. In our revised manuscript, we have also experimentally rule out the possibility of skyrmion motion by thermal gradients or local heats. The experimental observations and conclusions are not affected by the limitations of the model. We have now provided detailed information on the experiments, analytical model, and micromagnetic simulation codes in the paper.

Supplementary Fig. S13 | The measured out-of-plane magnetic hysteresis loop of Ta/Co₂₀Fe₆₀B₂₀/MgO/Ta by a vibrating sample magnetometer.

[52] Gueye, M., Zighem, F., Belmeguenai, M., Gabor, M. S., Tiusan, C., Faurie D. Spectroscopic investigation of elastic and magnetoelastic properties of CoFeB thin films. *J. Phys. D: Appl. Phys.* 49, 145003 (2016).

For clarity, the contents are provided in SUPPLEMENTARY INFORMATION page 12, line 19 and highlighted in yellow,

“

Supplementary Fig. S13 | The measured out-of-plane magnetic hysteresis loop of Ta/Co₂₀Fe₆₀B₂₀/MgO/Ta by a vibrating sample magnetometer.”

REVIEWER 2 COMMENTS:

Reviewer comments: The authors give detail explanation about the existence of Rayleigh waves in their samples and careful discussion about the influence of thermal effects on the skyrmion motion. The experimental results are clear and interesting; however, there are some minor issues should be clarified before it can be considered for publication.

Reply: We would like to express our gratitude to this reviewer for her/his valuable comments. We have carefully considered the reviewer's suggestions and have made necessary changes. These changes can now be found in the revised manuscript and supplementary materials. We believe that these revisions have strengthened the quality and clarity of our work. We are expecting that this reviewer can now support the publication of our manuscript. Below are our responses in a point-by-point manner.

Reviewer comments: 1. The authors claim that “the Rayleigh wave is capable of moving skyrmions only for a distance of less than half an acoustic wavelength, after which the skyrmions become trapped at the maximum particle displacement of the Rayleigh wave”. While, the velocity of SAWs propagating along piezoelectric substrates is always much larger than that of skyrmions. It does not seem to be reasonable that skyrmion could be trapped at the maximum particle displacement or other position in the wave. If so, a velocity of skyrmion motion up to thousand meter per second would be achieved. Authors need to clarify this statement and provide some details about the simulation.

Reply: We thank this reviewer for pointing out these important questions. We apologize for any confusion caused by our statement. Below are our specific responses:

It is true that the velocity of a SAW propagating along piezoelectric substrates is much larger than that of a skyrmion, the strain at the maximum particle displacement position is near zero. Additionally, the longitudinal displacement component of a Rayleigh is $\pi/2$ phase behind the vertical displacement component of a Rayleigh. As a result, the effects induced by the longitudinal displacement component and the vertical displacement component on a skyrmion can sometimes be opposite, resulting in a weak in-plane effective force that is insufficient to move the skyrmion significantly. To support this, we have performed additional simulation results that exhibit the position changes of a skyrmion under the action of a Rayleigh wave. These results can be found

in Supplementary Fig. S4. As shown in Fig. S4, the moving distance of a skyrmion under a Rayleigh wave is much smaller compared to that under an SH wave.

Furthermore, according to the reference [37], the skyrmions are created at the position where the effective torque is maximum, corresponding to the maximum particle displacement. Therefore, in our experiments, skyrmions created by Rayleigh waves do not exhibit significant movement.

We appreciate the reviewer for the thought-provoking idea regarding the potential for improving the skyrmion velocity. It is possible that the skyrmion velocity would improve under the action of acoustic waves with varying wavelengths generated by chirped interdigital transducers (IDTs) with electrode pitch variations or other specially designed IDTs. We will consider this possibility for further investigation in future studies.

Supplementary Fig. S4 | The simulated position changes of skyrmions under a Rayleigh wave and an SH wave. a, The position changes of a skyrmion under a Rayleigh wave. **b,** The position changes of a skyrmion under an SH wave.

[37] Yokouchi, T., Sugimoto, S., Rana, B. et al. Creation of magnetic skyrmion by surface acoustic waves. *Nat. Nanotechnol.* 15, 361-366 (2020).

For clarity, the contents are modified in page 1, line 51 right column and highlighted in yellow,

“We observe that a skyrmion does not exhibit significant movement under the action of a Rayleigh wave.”

Contents are modified in SUPPLEMENTARY INFORMATION page 6, line 6 and highlighted in yellow,

“Fig. S4 illustrates the position changes of skyrmions under the action of a Rayleigh wave and an SH wave, respectively. The moving distance of a skyrmion under a Rayleigh wave is smaller compared to that under an SH wave.”

Supplementary Fig. S4 | The simulated position changes of skyrmions under a Rayleigh wave and an SH wave. a, The position changes of a skyrmion under a Rayleigh wave. b, The position changes of a skyrmion under an SH wave.”

Reviewer comments: 2. The authors should give the SAW expression in their simulation like their previous work [J. Phys. D: Appl. Phys. 56, 084002 (2023)], an analytical model or a phenomenological model would be helpful for understanding.

Reply: Thank you for this valuable suggestion. In fact, it is important to provide a clear expression for the SAW, which can now be found as follows

$$\mathbf{u}_{SH} = \begin{bmatrix} u_x \\ u_y \\ u_z \end{bmatrix} = \begin{bmatrix} 0 \\ A \sin(kx - \omega t) \\ 0 \end{bmatrix}$$

where A , ω , and t are the amplitude of a SAW, the angular frequency of a SAW, and time, respectively. $k=2\pi/\lambda_{SAW}$ is the wavenumber of a SAW. λ_{SAW} is the wavelength of a SAW. The Rayleigh wave expression is defined as follows

$$\mathbf{u}_{Rayleigh} = \begin{bmatrix} u_x \\ u_y \\ u_z \end{bmatrix} = \begin{bmatrix} A_1 \sin(kx - \omega t) \\ 0 \\ A_2 \sin(kx - \omega t) \end{bmatrix}$$

We have now included these expressions in the SUPPLEMENTARY INFORMATION section, specifically on page 5, line 11, and highlighted them in yellow,

“The SH wave expression in the simulation is defined as follows

$$\mathbf{u}_{SH} = \begin{bmatrix} u_x \\ u_y \\ u_z \end{bmatrix} = \begin{bmatrix} 0 \\ A \sin(kx - \omega t) \\ 0 \end{bmatrix} \quad (S3)$$

where A , ω , and t are the amplitude of a SAW, the angular frequency of a SAW, and time, respectively. $k=2\pi/\lambda_{SAW}$ is the wavenumber of a SAW. λ_{SAW} is the wavelength of a SAW. The Rayleigh wave expression is defined as follows

$$\mathbf{u}_{Rayleigh} = \begin{bmatrix} u_x \\ u_y \\ u_z \end{bmatrix} = \begin{bmatrix} A_1 \sin(kx - \omega t) \\ 0 \\ A_2 \sin(kx - \omega t) \end{bmatrix} \quad (S4)$$

”

Reviewer comments: 3. The authors mention that “the dominant displacement components of Rayleigh waves are quite different depending on different cut types of piezoelectric substrates”, a reference is necessary to support this statement.

Reply: We apologize for missing this piece of information. We have included reference [5] in the SUPPLEMENTARY INFORMATION section to support this statement. Reference [5] investigates the variation of displacement components with the rotating angles of LiNbO₃.

For clarity, the contents are modified in SUPPLEMENTARY INFORMATION page 5, line 43 and highlighted in yellow,

“The dominant displacement components of Rayleigh waves are different depending on different cut types of piezoelectric substrates⁵.”

[5] Yamanouchi, K., Shibayama, K. Propagation and Amplification of Rayleigh Waves and Piezoelectric Leaky Surface Waves in LiNbO₃. J. Appl. Phys. 43, 856–862 (1972).

Reviewers' Comments:

Reviewer #1:

Remarks to the Author:

For NCOMM paper I expect a certain level of quality. Unfortunately, this paper does not strike me as an example of quality work. Theoretical analysis seems to be very superficial. My current critics:

1. I do not understand how $1c$ was obtained. If I simply plug Neel skyrmion ansatz in the magnetoelastic energy I get a different plot. If the plot also includes deformation of skyrmion due to strain, then I would check your parameters since magnetoelastic coupling is supposed to be a perturbation and should not substantially deform skyrmion. Could you plot a skyrmion in the presence of strain?

2. Related to question 1. If I use a Neel skyrmion ansatz and calculate a force on it from gradient of shear strain, I get a zero force from symmetry. Can it be that in your experiment the helicity of skyrmion is changed and it is not a Neel skyrmion? This could happen due to dipolar interactions.

Reviewer #2:

Remarks to the Author:

I am happy with the detailed response given to the questions raised in the last round and I think the work is ready to be published.

Response to Reviewers

We appreciate the reviewers for their valuable comments, which have helped us to further improve both the quality and the clarity of our work. Stimulated by their insightful comments, we have now revised our manuscript accordingly. Note that a revised manuscript that contains editing markups is also submitted. Below are our responses in a point-by-point manner.

REVIEWER 1 COMMENTS:

Reviewer comments: For NCOMM paper I expect a certain level of quality. Unfortunately, this paper does not strike me as an example of quality work. Theoretical analysis seems to be very superficial. My current critics:

Reply: We particularly thank this referee for a critical reading of our manuscript. We believe that the major concern of this referee arises from the “discrepancy” between the beautiful work done by Nepal, R., Güngördü, U., Kovalev, A. A. Appl. Phys. Lett. 112, 112404 (2018) and our micromagnetic simulation results. Stimulated by her/his specific concerns, we have now carefully examined our simulation to see the major difference between our simulation and the work by Nepal R., et al.

In short, we have found that such a “discrepancy” could be attributed to length scale of skyrmions in comparison with the SAW wavelength. In the case of a large ratio of the SAW wavelength to the skyrmion diameter, the strain gradient induced by a SAW is uniform across the skyrmion. Under this condition, the analytical conclusion of APL is absolutely correct, which has now been consistently verified by our simulation. In the case of a small ratio of the SAW wavelength to the skyrmion diameter, the strain gradient is non-uniform across the skyrmion, which leads to an asymmetric magnetoelastic energy density distribution. Under this condition, a larger net effective force acting on skyrmions could lead to directional transportation of skyrmions, which is being both numerically and experimentally verified by our results. This aspect has been adequately discussed in this revised manuscript. We believe that these aspects could clarify the concern regarding to the robustness of our manuscript. We are therefore hoping that this referee could be in the position of supporting the publication of our manuscript in Nature Communications.

Reviewer comments: 1. I do not understand how $1c$ was obtained. If I simply plug Neel skyrmion ansatz in the magnetoelastic energy I get a different plot. If the plot also includes deformation of skyrmion due to strain, then I would check your parameters since magnetoelastic coupling is supposed to be a perturbation and should not substantially deform skyrmion. Could you plot a skyrmion in the presence of strain?

Reply: We thank this referee for these intriguing comments. Fig. 1c is obtained by using MuMax3 with a magnetoelastic coupling solver. Detailed description of which can now be found in the page 6 of the manuscript and page 3 of the Supplementary Information. In the micromagnetic simulation, we have found that the magnetoelastic energy density distribution strongly depends on the relative size of the skyrmion in comparison to the wavelength of the SAW. When the size of the skyrmion is comparable with the wavelength of the SAW, the strain gradient induced by a SAW becomes non-uniform across the skyrmion, resulting in an asymmetric magnetoelastic energy density distribution, as shown in Fig. 1c (with a ratio of the SAW wavelength to the skyrmion diameter $\frac{\lambda}{D_{sk}} = 8$).

To systematically address this specific concern, we have now extended our micromagnetic simulations with a much large SAW wavelength ($\frac{\lambda}{D_{sk}} = 800$), where the strain gradient induced by a SAW can be considered to be uniform on the skyrmion. We found a nearly symmetric magnetoelastic energy density distribution, as shown in Fig. R1. Note that this result is consistent with the theoretical prediction by Nepal, R., Güngördü, U., Kovalev, A. A. *Appl. Phys. Lett.* 112, 112404 (2018).

Figure R1. The magnetoelastic energy density distribution of a skyrmion under an elastic wave with periodic shear vertical displacements when the ratio of the SAW wavelength to the skyrmion diameter is 800.

We have also provided the simulation of a skyrmion in the presence of strain as shown in Supplementary Fig. S5. We found that there is no appreciable deformation of the skyrmion by using the same simulation parameters. We acknowledge that skyrmions can become unstable and deform when subjected to significant strain, consistent with previous experimental findings [13].

Supplementary Fig. S5 | The simulated skyrmion under an elastic wave with periodic shear vertical displacements.

[13] Ba, Y., Zhuang, S., Zhang, Y. et al. Electric-field control of skyrmion in multiferroic heterostructure via magnetoelectric coupling. Nat. Commun. 12, 1-10 (2021).

For clarity, the contents are modified in page 1, line 55 right column and highlighted in yellow,

“The magnetoelastic energy density distribution of a skyrmion strongly depends on the relative size of the skyrmion in comparison to the wavelength of the SAW. When the size of the skyrmion is comparable with the wavelength of a SAW, the strain gradient induced by a SAW becomes non-uniform across the skyrmion, resulting in an asymmetric magnetoelastic energy density distribution.”

The contents are provided in SUPPLEMENTARY INFORMATION page 7, line 1 and highlighted in yellow,

“

Supplementary Fig. S5 | The simulated skyrmion under an elastic wave with periodic shear vertical displacements.”

Reviewer comments: 2. Related to question 1. If I use a Neel skyrmion ansatz and calculate a force on it from gradient of sheer strain, I get a zero force from symmetry. Can it be that in your experiment the helicity of skyrmion is changed and it is not a Neel skyrmion? This could happen due to dipolar interactions.

Reply: We greatly thank the reviewer for pointing out this intriguing question. We fully agree with the reviewer that, according to the analytical model in ref.[38], the net force on a Néel skyrmion should be zero under the uniform shear strain gradient. This can also be verified by our micromagnetic simulation when we set the wavelength of a SAW much larger than the diameter of the skyrmion. This aspect has now been discussed in Supplementary Fig. S6a, which shows the simulated magnetoelastic force density distribution with the ratio of the SAW wavelength to the skyrmion diameter ($\frac{\lambda}{D_{sk}}$) = 800. In this case, the magnetoelastic force density distribution of the skyrmion is nearly symmetric. And the net magnetoelastic force acting on the skyrmion can be calculated in the order of 3×10^{-2} nN. By continuously decreasing the wavelength of a SAW (and hence the ratio of $\frac{\lambda}{D_{sk}}$), the shear strain gradient across the skyrmion becomes non-uniform which induces a larger net effective magnetoelastic force, as

shown in Supplementary Fig. S6b. In our experiments, $\frac{\lambda}{D_{sk}}$ was set to be 8, and the simulated effective magnetoelastic force is calculated to be 1.9 nN. This is almost two orders of magnitudes larger than that with $\frac{\lambda}{D_{sk}}=800$ (3×10^{-2} nN).

With respect to the helicity reversal. Typically, in the magnetic multilayers made of heavy metal/ultrathin ferromagnets, the helicity is determined by the sign of interfacial DMI. Once the film is deposited, the sign of interfacial DMI is determined, which leads to a fixed helicity of spin textures. As of the present material system, Ta/CoFeB/MgO, it is known to host left-handed. This part has been well documented from the existing literatures [10],[20].

On the other hand, to fully rule out the concern of this referee, we have performed detailed micromagnetic simulations to reveal the evolution of helicity of skyrmions as a function of strains induced by SAWs, as shown in Fig. R2. In particular, we have identified that the helicity of skyrmion by using the same material parameter, remains the same as left-handed.

Supplementary Fig. S6 | The simulation of the magnetoelastic force. **a**, The simulated magnetoelastic force density distribution of a skyrmion under an SH wave when the ratio of the SAW wavelength to the skyrmion diameter $\frac{\lambda}{D_{sk}} = 800$. **b**, The net effective magnetoelastic force on the skyrmion as a function of $\frac{\lambda}{D_{sk}}$.

Figure R2. The simulated skyrmions under SAWs. **a**, the strain amplitude induced by a SAW is 0.026. **b**, the strain amplitude induced by a SAW is 2.6×10^{-3} . **c**, the strain amplitude induced by a SAW is 2.6×10^{-4} .

[10] Woo, S., Litzius, K., Krüger, B. et al. Observation of room-temperature magnetic skyrmion and their current-driven dynamics in ultrathin metallic ferromagnets. Nat. Materials, 15, 501-506 (2016).

[20] Wang, Z., Guo, M., Zhou, HA. et al. Thermal generation, manipulation and thermoelectric detection of skyrmion. Nat. Electron. 3, 672-679 (2020).

[38] Nepal, R., Güngördü, U., Kovalev, A. A. Magnetic skyrmion bubble motion driven by surface acoustic waves. Appl. Phys. Lett. 112, 112404 (2018).

For clarity, the contents are modified in page 2, line 16 left column and highlighted in yellow,

“When we set the wavelength of a SAW much larger than the diameter of the skyrmion, the strain gradient induced by a SAW can be considered to be uniform across the skyrmion. In this case, the magnetoelastic force density distribution of a skyrmion is nearly symmetric. And the net magnetoelastic force on a skyrmion approaches zero. This has also been predicted by the analytical model in the early report³⁸.”

The contents are provided in SUPPLEMENTARY INFORMATION page 7, line 10 and highlighted in yellow,

“Supplementary Fig. S6a shows the simulated magnetoelastic force density distribution of a skyrmion with the ratio of the SAW wavelength to the skyrmion diameter $\frac{\lambda}{D_{sk}} = 800$. In this case, the magnetoelastic force density distribution of the skyrmion is nearly symmetric. And the net magnetoelastic force on the skyrmion can be calculated in the order of 3×10^{-2} nN. By continuously decreasing the wavelength of a SAW (and hence the ratio of $\frac{\lambda}{D_{sk}}$), the shear strain gradient across the skyrmion becomes non-uniform which induces a larger net effective magnetoelastic force, as shown in Supplementary Fig. S6b.

Supplementary Fig. S6 | The simulation of the magnetoelastic force. a, The simulated magnetoelastic force density distribution of a skyrmion under an SH wave when the ratio of the SAW wavelength to the skyrmion diameter $\frac{\lambda}{D_{sk}} = 800$. **b,** The net effective magnetoelastic force on the skyrmion as a function of $\frac{\lambda}{D_{sk}}$.”

REVIEWER 2 COMMENTS:

Reviewer comments: I am happy with the detailed response given to the questions raised in the last round and I think the work is ready to be published.

Reply: We would like to express our gratitude to this reviewer for supporting the publication of our work.

Reviewers' Comments:

Reviewer #1:

Remarks to the Author:

I am happy with the response, and I believe the paper is ready for publication.

Response to Reviewers

We appreciate the reviewers for their valuable comments, which have helped us to further improve both the quality and the clarity of our work. Below are our responses in a point-by-point manner.

REVIEWER 1 COMMENTS:

Reviewer comments: I am happy with the response, and I believe the paper is ready for publication.

Reply: We would like to express our gratitude to this reviewer for supporting the publication of our work. Thank you very much.